# Excitatory synapses and gap junctions cooperate to improve Pv neuronal burst firing and cortical social cognition in *Shank2*-mutant mice

Eunee Lee [1,2,10], Seungjoon Lee[3,10], Jae Jin Shin [4,5,10], Woochul Choi[6,10], Changuk Chung[1], Suho Lee [3], Jihye Kim [1], Seungmin Ha[1], Ryunhee Kim[1], Taesun Yoo[1], Ye-Eun Yoo[3], Jisoo Kim[3], Young Woo Noh [3], Issac Rhim[1], Soo Yeon Lee[3], Woohyun Kim[3], Taekyung Lee[1], Hyogeun Shin[7], Il-Joo Cho [7], Karl Deisseroth [8], Sang Jeong Kim[9], Joo Min Park[5✉], Min Whan Jung [1,3✉], Se-Bum Paik [6✉] & Eunjoon Kim [1,3✉]

NMDA receptor (NMDAR) and GABA neuronal dysfunctions are observed in animal models of autism spectrum disorders, but how these dysfunctions impair social cognition and behavior remains unclear. We report here that NMDARs in cortical parvalbumin (Pv)-positive interneurons cooperate with gap junctions to promote high-frequency (>80 Hz) Pv neuronal burst firing and social cognition. *Shank2*$^{-/-}$ mice, displaying improved sociability upon NMDAR activation, show impaired cortical social representation and inhibitory neuronal burst firing. Cortical *Shank2*$^{-/-}$ Pv neurons show decreased NMDAR activity, which suppresses the cooperation between NMDARs and gap junctions (GJs) for normal burst firing. *Shank2*$^{-/-}$ Pv neurons show compensatory increases in GJ activity that are not sufficient for social rescue. However, optogenetic boosting of Pv neuronal bursts, requiring GJs, rescues cortical social cognition in *Shank2*$^{-/-}$ mice, similar to the NMDAR-dependent social rescue. Therefore, NMDARs and gap junctions cooperate to promote cortical Pv neuronal bursts and social cognition.

[1] Center for Synaptic Brain Dysfunctions, Institute for Basic Science (IBS), Daejeon, Korea. [2] Department of Anatomy, College of Medicine, Yonsei University, Seoul, Korea. [3] Department of Biological Sciences, KAIST, Daejeon, Korea. [4] Department of Brain and Cognitive Science, College of Natural Science, Seoul National University, Seoul, Korea. [5] Center for Cognition and Sociality, Institute for Basic Science (IBS), Daejeon, Korea. [6] Program of Brain and Cognitive Engineering, Department of Bio and Brain Engineering, Korea Advanced Institute for Science and Technology (KAIST), Daejeon, Korea. [7] Center for BioMicrosystems, Brain Science Institute, Korea Institute of Science and Technology (KIST), Seoul, Korea. [8] Department of Bioengineering, Department of Psychiatry and Behavioral Sciences, Howard Hughes Medical Institute, Stanford University, Stanford, CA, USA. [9] Department of Physiology, College of Medicine, Seoul National University, Seoul, Korea. [10] These authors contributed equally: Eunee Lee, Seungjoon Lee, Jae Jin Shin, Woochul Choi.
✉email: joominp@ibs.re.kr; mwjung@kaist.ac.kr; sbpaik@kaist.ac.kr; kime@kaist.ac.kr

Autism spectrum disorders (ASD) represent a major neuropsychiatric disorder characterized by social deficits and repetitive behaviors for which a large number of neurobiological mechanisms have been suggested[1–10]. NMDA receptor (NMDAR) dysfunctions have been suggested to underlie ASD-related phenotypes in many animal models of ASD[11–16], although related brain regions, cell types, and synaptic and circuit mechanisms remain largely unclear. GABA neuronal dysfunctions have also been increasingly implicated in ASD pathophysiology[1,17–23], although specific cell types and mechanisms remain largely unclear. Previous studies on parvalbumin (Pv)-positive neurons, a major GABAergic neuron type[24], reported key roles of Pv neurons in the regulation of social cognition and behaviors[21,25,26]. Currently, whether and how the abovementioned ASD-related NMDAR dysfunctions are associated with GABA or Pv neuronal dysfunctions and social deficits have remained largely obscure[27–30].

Previous studies on GABA neuronal dysfunctions in ASD mainly focused on well-known functions of Pv neurons such as synaptic inhibition of pyramidal neurons (PNs) and regulation of local gamma oscillations. Here, we focused on largely neglected aspects of Pv neurons in ASD, namely gap junctions and burst firing. Gap junctions electrically couple different GABA neurons to promote their biochemical and electrical communications, playing important roles in concerted neuronal activation and burst firings in health and diseases (brain injury, seizure, and Alzheimer's disease)[31–34]. Neuronal burst firings, defined as short high-frequency trains of action potentials, are thought to promote the reliability of information transfer between neurons, encode parallel stimulus features, prepare target neurons for subsequent inputs, and signal coincident neural processes[35–41]. Neuronal bursts have been associated with various brain functions and dysfunctions, including sensory perception, reward, epilepsy, and depression[42–45]. Hence, alterations in Pv neuronal gap junctions and burst firing could have profound impacts on diverse cortical functions including social cognition. However, whether and how Pv neuronal GAP junctions and burst firing contribute to social cognition, and whether and how their dysfunctions are related to ASD are not well known. In fact, even such simple issues as to whether Pv neurons display burst firing and how Pv neuronal bursts are regulated by gap junctions are not known, although a previous study has shown that Pv-positive multipolar bursting cells in the mouse neocortex, distinct from fast-spiking Pv neurons, generate bursts that are regulated by muscarinic signaling[46].

In the present study, we investigated how Pv neuronal gap junctions and burst firing are related to social cognition in a mouse model of ASD. We also investigated how they are related to NMADR in supporting social cognition. The results suggest that NMDARs and gap junctions cooperate to promote cortical Pv neuronal bursts and social cognition.

## Results

### Abnormal cortical social representation in *Shank2*⁻/⁻ mice.

To explore cortical mechanisms underlying ASD-related social deficits, we attempted to measure and compare neuronal firings in live wild-type (WT) and *Shank2*⁻/⁻ mice, a mouse model of ASD[15], engaged in social interaction. We targeted the medial prefrontal cortex (mPFC), a brain region with strong social implications[25,47–50].

*Shank2*⁻/⁻ or WT mice with tetrodes implanted in the mPFC (5 pairs; AP + 1.7 mm, ML ± 0.3 mm) (Supplementary Fig. 1a) were placed in a rest area for 5 min and moved to a linear apparatus with a narrow corridor connected to with two empty side chambers for 10-min free exploration (E [empty]–E session) (Fig. 1a)[48]. Social and non-social (inanimate object) targets were

then placed in the side chambers. After 10-min exploration (first S [social]–O [object] session), targets were switched (second S–O session; or O–S session) to dissociate target-dependent firing from location-dependent firing[51].

*Shank2*⁻/⁻ mice showed decreased social interaction in the first S–O session, compared with WT mice, although the difference became weak in the second S–O session likely attributable to social habituation in WT mice (Supplementary Fig. 1c). However, discriminative neuronal responses in the second S–O session, as indicated by the discrimination index (*d′*; differential neuronal responses for two targets)[48,52], were significantly greater than those in the E–E session in *Shank2*⁻/⁻ mice and tended to be greater in WT mice (Supplementary Fig. 1d), suggestive of normal cognition of social and non-social targets. In addition, our previous study on WT mice reported stronger cortical discriminative neuronal responses in both first and second S–O sessions relative to the E–E session[48]. We thus pooled neuronal recordings from both S–O sessions for further analyses.

We then first determined target-specific firing of mPFC neurons by comparing neuronal responses to S, O, and E targets. Some neurons showed single target-specific firing (Fig. 1b [left] and Supplementary Fig. 1b; i.e., S-specific firing) while others showed broadly-tuned firing (Fig. 1b [right] and Supplementary Fig. 1b) in both WT and *Shank2*⁻/⁻ mice.

We next plotted the recorded neurons based on their relative probabilities of discriminative firing for S vs. O targets and also for S vs. E targets in a two-by-two matrix to identify the neurons that show 'target-discriminative' firings (i.e., S-discriminative neurons responding more strongly or weakly for S over O) and 'target-specific' firings (i.e., S-specific neurons being derived from the combination of S–O and S–E discriminative neurons) (Supplementary Fig. 2).

These target-discriminative and target-selective neurons were plotted in Venn diagrams (Fig. 1c) and used to compare the differences between WT and *Shank2*⁻/⁻ mice. Intriguingly, there was no genotype difference in the proportion of social-specific mPFC neurons. However, proportions of non-social target (E, O, and sideness/Si)-specific neurons were increased in *Shank2*⁻/⁻ mice (Fig. 1d).

In addition, when the total number of target/sideness pairs (S–E, S–O, O–E, and L–R [left–right; sideness]) that a neuron can discriminate was calculated based on the Venn-diagram results (i.e., the neurons in the center of the Venn diagram and inside the parenthesis can discriminate a total of four target pairs (S–E, S–O, O–E, and L–R)), there was an overall increase in the total number of target pairs in *Shank2*⁻/⁻ mice relative to WT mice (Fig. 1e).

These differences in firing patterns between WT and *Shank2*⁻/⁻ mice are less likely to be attributable to undersampling in *Shank2*⁻/⁻ mice because the total numbers of sniffing (nose-poke) trials for E–E, S–O, and O–S sessions were comparable in WT and *Shank2*⁻/⁻ mice (genotype $p = 0.248$; two-way ANOVA). These results collectively suggest that non-social target-discriminative neurons are more prevalent in the *Shank2*⁻/⁻ mPFC relative to social target-discriminative neurons.

### Impaired burst firing in *Shank2*⁻/⁻ mPFC inhibitory neurons.

To explore further how mPFC neuronal firing differs between WT and *Shank2*⁻/⁻ mice, we divided the recorded neurons into putative excitatory and inhibitory (pExc and pInh) neurons based on physiological characteristics (Fig. 2a), and examined their discharge rates and spiking patterns. pExc, but not pInh, neurons elevated firing rates upon exposure to S/O/E targets relative to the R (rest) state in both WT and *Shank2*⁻/⁻ mice (Fig. 2b). A difference, however, was found in high-frequency burst firing, defined as those with inter-spike intervals (ISIs) <12 ms

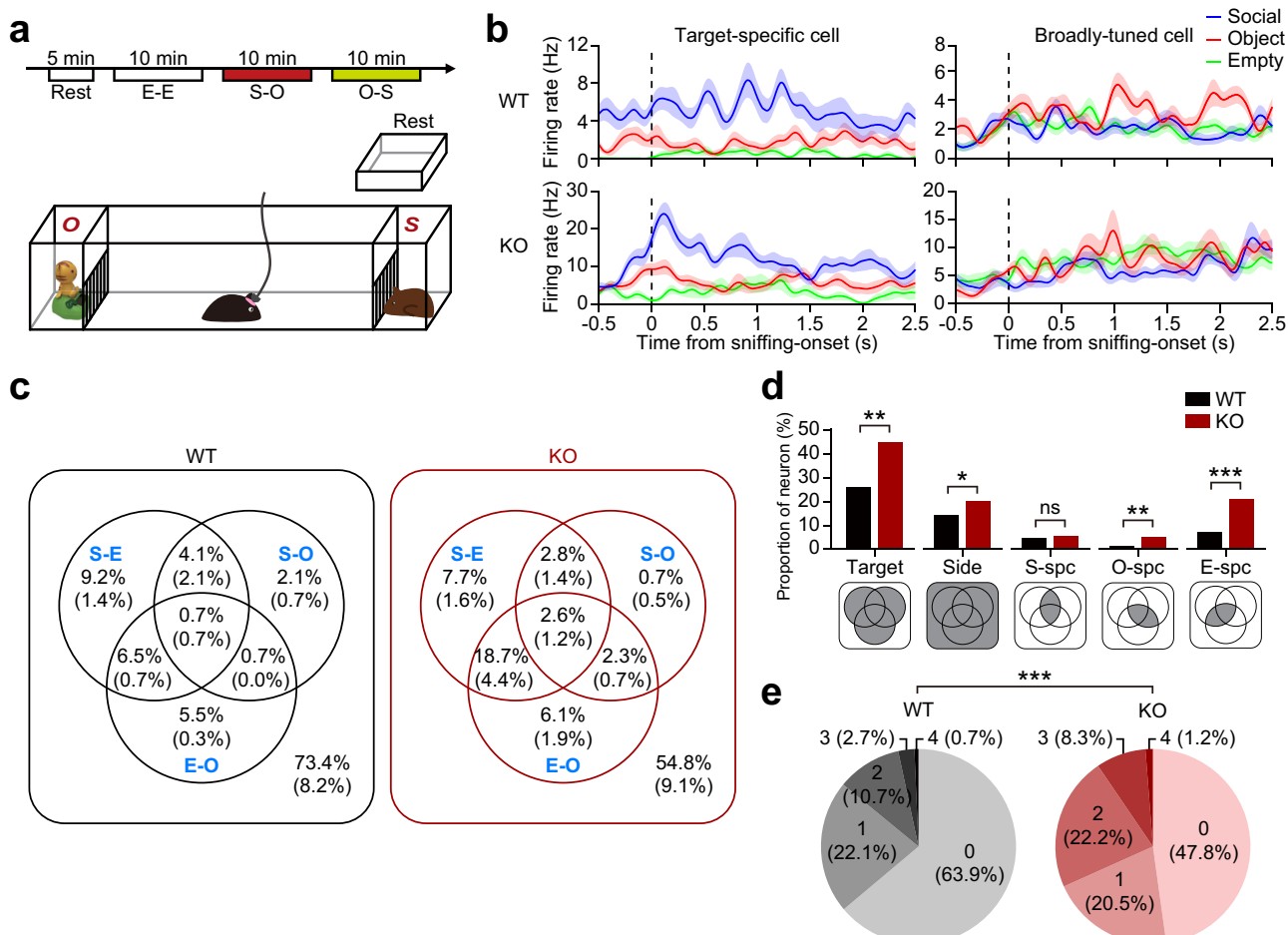

**Fig. 1 Abnormal social representation in the *Shank2⁻/⁻* mPFC. a** Schematic diagram showing the linear apparatus for single-unit recordings in the mPFC of mice exploring social/non-social targets. The experiments included a 5-min rest period and three 10-min sessions (E–E, 1st S–O [S–O], and 2nd S–[O–S]). E/S/O, empty/social/object target. **b** Sample mPFC neurons showing single target-specific and broadly-tuned responses in WT and *Shank2⁻/⁻* mice. Shown are spike density functions ($\sigma = 6.2$ ms) aligned to immediately before and after target (E/O/S) sniffing (nose-poke). Data: mean ± SEM. **c** Discharge patterns of mPFC neurons to social and non-social targets in WT and *Shank2⁻/⁻* mice (>12 weeks) summarized in Venn diagrams, based on their discriminative discharges for S–E, S–O, and E–O target combinations (see Supplementary Fig. 2). Numbers in the Venn diagrams indicate proportions (%) of neurons that are either target-discriminative (i.e., S–O-discriminative neurons showing higher or lower firing for S over O) or target-selective (i.e., S-specific neurons being derived from the combination of S–O- and S–E-discriminative neurons). The neurons in the parentheses are additionally discriminative to sidedness/Side (left/right side of the apparatus) ($n = 295$ neurons/5 mice [WT], 409/5 mice [KO]). **d** Proportions (%) of the neurons that are target-specific (S/O/E/Side/Target [all targets]) in the WT and *Shank2⁻/⁻* mPFC derived from the Venn-diagram results (**c**) corresponding to the shaded miniature Venn-diagram areas (**d**). Note that 'Target' neuronal proportion is not the sum of the other target-selective neurons (S/O/E/Side), as explained in the shaded Venn diagrams ($n = 295/5$ mice [WT] 409/5 mice [KO], *$p < 0.05$, **$p < 0.001$, ***$p < 0.001$, ns, not significant, Chi-square test; $p = 0.0021$ [Target], $p = 0.0194$ [Side], $p = 0.549$ [S-specific], $p = 0.007$ [O-specific], $p < 0.0001$ [E-specific]). **e** Total numbers and proportions of target pairs that one neuron can discriminate (S–O, S–E, E–O, and L–R [left–right]) based on the Venn-diagram summary (**c**) (i.e., a neuron in the center of the Venn diagram within the parenthesis can discriminates four target pairs (S–E, S–O, E–O, and L–R) ($n = 295$ [WT] and 409 [KO], ***$p < 0.001$, Chi-square test). See Source data for raw data values and Supplementary Table 1 for statistical details.

(corresponding to >80 Hz), between WT and *Shank2⁻/⁻* pInh neurons (Fig. 2c). This definition of burst firing was based on the peak ISI of ~13 ms, and each burst firing contained, in large parts (~80%), two spikes (Supplementary Fig. 3a, b).

We compared how burst firing changes during S/O/E target encounter relative to the R state (Δburst firing) between WT and *Shank2⁻/⁻* mice. There was a significant interaction between the effects of target and genotype on Δburst firing of pInh neurons (two-way ANOVA, $p = 0.008$). Δburst firing was significantly larger for S than O and E targets in WT mice (Sidak's test, $p = 0.009$ and 0.032, respectively), but not significant across targets in *Shank2⁻/⁻* mice ($p > 0.05$) (Fig. 2d). In addition, Δburst firing for S target differed significantly between WT and *Shank2⁻/⁻* mice ($p = 0.001$). No significant

effect of target, genotype, or their interaction was found in Δburst firing of pExc neurons (genotype, $p = 0.249$; target, $p = 0.515$; genotype × target interaction, $p = 0.450$). These differences were maintained by broader definitions of burst firing (ISI of 6 ms and 24 ms instead of 12 ms) (Supplementary Fig. 3c, d) but eliminated by shuffling spike times (Supplementary Fig. 3e), suggesting that firing rates in pInh neurons do not affect Δburst firing. These results suggest that pInh neurons fail to increase high-frequency burst firing upon social target exposure, even though mean activity changes of pExc and pInh neurons upon target exposure were similar in *Shank2⁻/⁻* mice. These changes in burst firing may involve both Pv and somatostatin (Som)-positive neurons because they comprise ~40% and 30% of neocortical interneurons, respectively[24], and

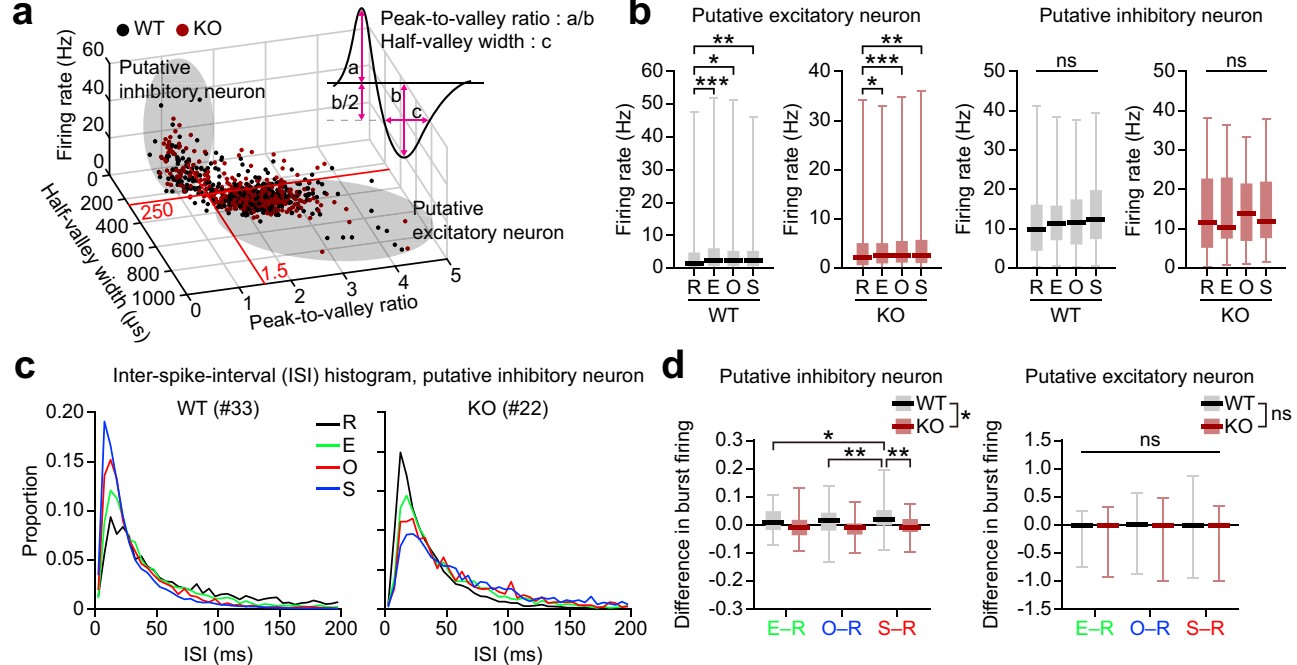

**Fig. 2 Impaired burst firing in *Shank2*$^{-/-}$ mPFC inhibitory neurons. a** Classification of mPFC neurons into putative excitatory and inhibitory neurons. Total neuronal firings, grouped into putative excitatory and inhibitory firings, based on peak-to-valley ratio and half-valley width. Total spikes from WT (black) and *Shank2*$^{-/-}$ (red) mice were plotted three-dimensionally against peak-to-valley ratio, half-valley width, and firing rate. Spikes were grouped into putative excitatory and inhibitory firings based on peak-to-valley ratio and half-valley width, as follows: excitatory firings, peak-to-valley ratio >1.5 and half-valley width >250 μs; inhibitory firings, peak-to-valley ratio <1.5, and half-valley width <250 μs. **b** Increased firing rates in pExc, but not pInh, neurons upon target encounters, as shown by mean-firing rate (±SEM) of pExc and pInh neurons of WT and *Shank2*$^{-/-}$ mice in the resting (R) state and with E/O/S encounters ($n = 295$ [WT] and 409 [KO], *$p < 0.05$, **$p < 0.01$, ***$p < 0.001$, Friedman test with Dunn's multiple comparison; $p = 0.0005$ [R–E], $p = 0.0005$ [R–O], $p = 0.0290$ [R–S], $p = 0.0072$ [WT-pExc], $p < 0.0001$ [R–E], $p = 0.0109$ [R–O], $p < 0.0001$ [R–S], $p = 0.0030$ [KO-pExc], $p = 0.258$ [WT-pInh], $p = 0.118$ [KO-pInh]). **c** ISI histograms for two sample pInh neurons (left, from WT; right, from KO mouse) in the resting (R) state and with E/O/S encounters. The numbers in the parenthesis indicate single-neuron numbers. **d** Bar graphs for Δburst firing (changes in high-frequency burst firings [<12 ms of ISI] during S/O/E target encounter relative to the R state) for E, O, and S in WT and KO-pInh neurons ($n = 36$ [WT] and 41 [KO] for pInh, 286 [WT] and 340 [KO] for pExc, *$p < 0.05$, **$p < 0.01$, two-way repeated-measures/RM-ANOVA with Sidak's multiple comparison test [bar graph]; genotype $p = 0.0129$, target $p = 0.2808$, interaction $p = 0.0082$ [pInh], E–R $p = 0.1945$, O–R $p = 0.350$, S–R $p = 0.0011$ [WT vs KO, pInh], E–R vs O–R $p = 0.95$, E–R vs S–R $p = 0.032$, O–R vs S–R $p = 0.0085$ [WT-pInh], E–R vs O–R $p = 0.997$, E–R vs S–R $p = 0.762$, O–R vs S–R $p = 0.650$ [KO-pInh], genotype $p = 0.249$, target $p = 0.515$, interaction $p = 0.450$ [pExc]). See Source data for raw data values and Supplementary Table 1 for statistical details.

~70% and ~30% of functional pInh neurons in the mPFC[53], and both types of cells in the prefrontal cortex are known to regulate cognitive and social functions[26,53–58].

**Abnormal oscillations in the *Shank2*$^{-/-}$ mPFC.** Cortical neurons, including Pv neurons, strongly affect local network activities, including gamma oscillations[59–63], and oscillatory dysfunctions have been associated with ASD[64–66]. We thus analyzed oscillatory activities in the *Shank2*$^{-/-}$ mPFC using local field potentials (LFPs) measured during the rest and target-encountering periods.

The total powers of rest-period oscillations were abnormally high in nearly all frequency ranges, including gamma, in the *Shank2*$^{-/-}$ mPFC, as compared with WT mice (Fig. 3a). Upon encountering (sniffing or nose-poke to) targets (S, O, and E), there were marked decreases in the oscillatory powers (data from S, O, and E sessions combined). The WT mPFC, however, did not show target-induced decreases in oscillatory powers, except for a moderate decrease in the theta range.

We next compared 'changes' in oscillatory powers immediately before (−1.5 to −0.5 s) and after (+0.5 to +1.5 s) target encounters in WT and *Shank2*$^{-/-}$ mice. In *Shank2*$^{-/-}$ mice, the powers for S, O, and E targets (z-scores relative to rest-period baselines) tended to decrease strongly, especially in the gamma

(30–150 Hz) and alpha (12–30 Hz) ranges, after sniffing (nose-poke) onset (Fig. 3b, c; Supplementary Fig. 4), contrary to the WT patterns where target encountering caused minimal decreases in the power. Therefore, *Shank2*$^{-/-}$ mPFC displays abnormally heightened resting-state oscillations that are markedly decreased in the gamma range by target encounters.

**_Shank2_$^{-/-}$ Pv neurons fail to inhibit target PNs.** To understand the mechanisms underlying the altered neuronal firings and local oscillations in the *Shank2*$^{-/-}$ mPFC, we measured synaptic transmission and neuronal excitability by slice recordings of mPFC pyramidal neurons (PNs) and Pv neurons (prelimbic region, layer 2/3) at late stages (>12 weeks) when Pv neurons are fully developed. We focused on Pv neurons among known interneuron types because (1) they represent the most abundant interneuron type[67], and (2) Pv neuronal density tended to decrease in the *Shank2*$^{-/-}$ mPFC (Supplementary Fig. 5), indicative of suppressed Pv neuronal differentiation or activity.

In *Shank2*$^{-/-}$ PNs, there were no changes in spontaneous synaptic transmission or neuronal excitability, as shown by spontaneous excitatory and inhibitory postsynaptic currents (sEPSCs and sIPSCs) and current-firing curve (Fig. 4a–c). *Shank2*$^{-/-}$ Pv neurons also showed unaltered sEPSCs, sIPSCs, and neuronal excitability (Fig. 4d–f). These results indicate

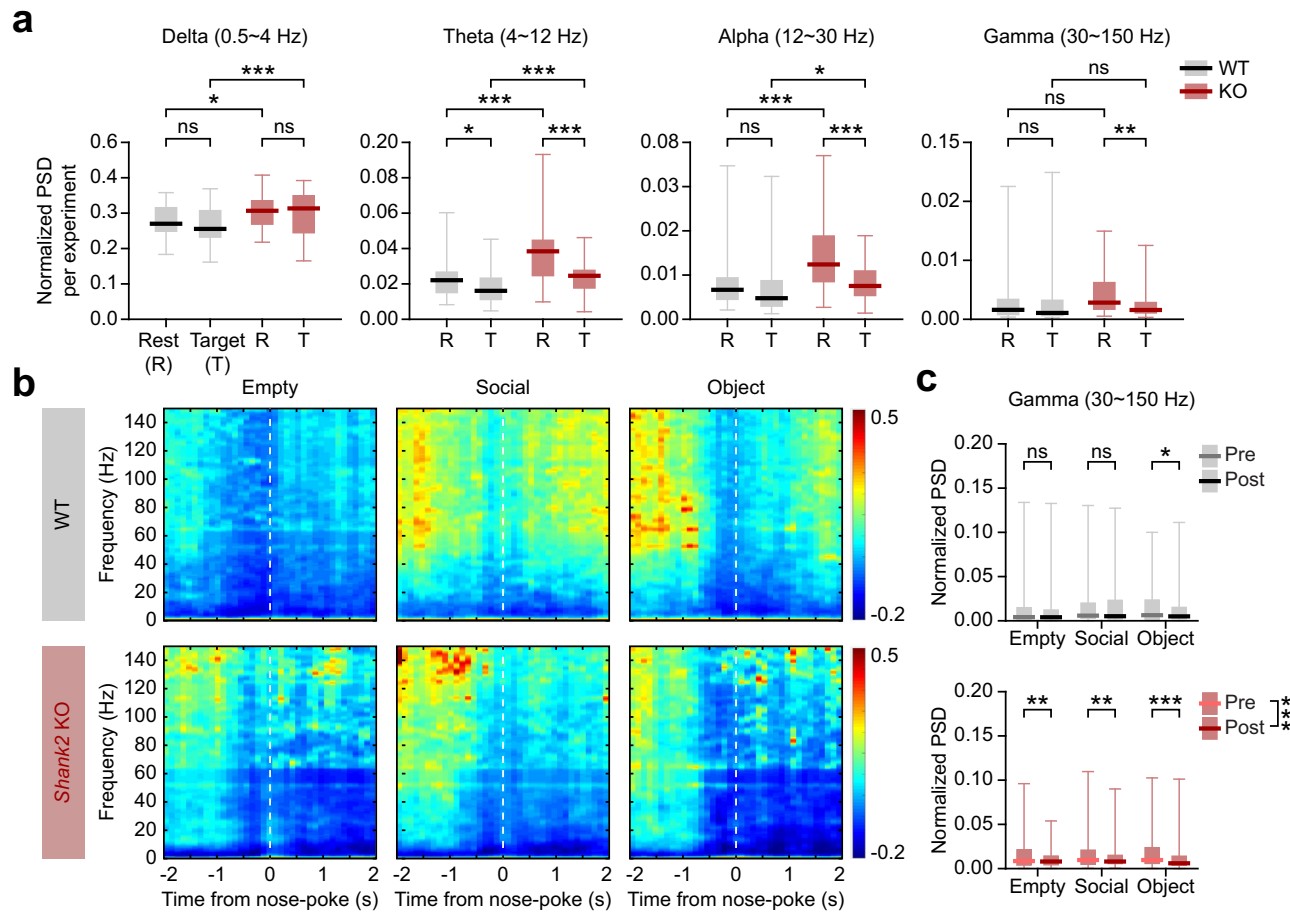

**Fig. 3 Enhanced resting-state, but suppressed target-induced, oscillations in the *Shank2⁻/⁻* mPFC. a** Enhanced resting-state, but suppressed target (S/O/E combined)-induced, oscillations in the *Shank2⁻/⁻* mPFC compared with WT mice. Local field potentials (LFPs) measured during rest periods and the entire period of S/O/E target exploration were used for the analysis (n = 53 experiments from 5 WT mice, and 52 experiments from 5 KO mice, *$p < 0.05$, **$p < 0.01$, ***$p < 0.001$, ns, not significant, two-way ANOVA with Tukey's test; genotype $p < 0.001$, event $p = 0.153$, interaction $p = 0.712$ [delta], rest $p = 0.014$, target WT-KO $p < 0.001$ [WT-KO-delta], WT $p > 0.05$, KO $p > 0.05$ [rest-target-delta], genotype $p < 0.001$, event $p < 0.001$, interaction $p < 0.001$ [theta], rest $p < 0.001$, target $p < 0.001$ [WT-KO-theta], WT $p = 0.014$, KO $p < 0.001$ [rest-target-theta], genotype $p < 0.001$, event $p < 0.001$, interaction $p = 0.002$ [alpha], rest $p < 0.001$, target $p = 0.039$ [WT-KO-alpha], WT $p = 0.331$, KO $p < 0.001$ [rest-target-alpha], genotype $p = 0.548$, event $p = 0.025$, interaction $p = 0.062$ [gamma], rest $p = 0.153$, target $p = 0.204$ [WT-KO-gamma], WT $p = 0.79$, KO $p = 0.004$ [rest-target-gamma]). **b** Temporal changes in color-coded z-scores for spectral powers before and after sniffing (nose-poking) of S/O/E targets in WT and *Shank2⁻/⁻* mice. Z-scores were derived by normalizing powers in the time period around sniffing (nose pokes) to those during the rest period (see the "Methods" section for details on the z-score). **c** Changes in power spectral densities (PSD) before and after sniffing (nose pokes; −1.5 to −0.5 s and 0.5 to 1.5 s, respectively). Results for the gamma range are shown (n = 52 experiments from 5 WT mice, and 53 experiments from 5 KO mice, *$p < 0.05$, **$p < 0.01$, ***$p < 0.001$, ns, not significant, two-way ANOVA with Tukey's test; time $p = 0.441$, target $p = 0.009$, interaction $p = 0.031$ [WT], pre $p = 0.045$, post $p = 0.003$ [E-S-WT], Empty $p = 0.677$, Social $p = 0.258$, object $p = 0.023$ [pre-post-WT], time $p < 0.001$, target $p = 0.149$, interaction $p = 0.346$ [KO], pre $p > 0.05$, post $p > 0.05$ [E-S-KO], Empty $p = 0.003$, Social $p = 0.005$, Object $p < 0.001$ [pre-post-KO]). See Source data for raw data values and Supplementary Table 1 for statistical details.

unaltered baseline synaptic transmission and neuronal excitability in *Shank2⁻/⁻* PNs and Pv neurons.

We next measured evoked inhibitory synaptic transmission in the Pv-to-PN axis using opto-patch experiments. To this end, we generated *Pv-Cre;Shank2⁻/⁻* mice by crossing *Shank2⁻/⁻* mice with *Pv-Cre* mice, which differs from *Pv-Cre;Shank2*^fl/fl mice with Pv neuron-specific Shank2 deletion reported previously[30]. These mice were injected with AAV-Dio-ChR2-EYFP in the mPFC to drive Pv neuron-specific ChR2 expression (Fig. 4g).

Light stimulation of WT Pv neurons in the prelimbic region of mPFC slices at 10 Hz (473 nm, 5 ms), a frequency that effectively inhibits target PNs for information processing[21], decreased the excitability of target PNs (Fig. 4h). In sharp contrast, *Shank2⁻/⁻* Pv neurons failed to inhibit target PNs (Fig. 4i), although the spike probability was moderately decreased immediately after the

5-ms light pulses (Fig. 4j, k). Therefore, stimulated *Shank2⁻/⁻* mPFC Pv neurons fail to inhibit PNs, although their baseline functions (i.e., spontaneous inhibitory synaptic transmission) are unaltered.

**Enhanced burst firing in *Shank2⁻/⁻* Pv neurons.** The impaired PN inhibition by stimulated *Shank2⁻/⁻* Pv neurons might be attributable to altered firing properties of Pv neurons. We thus attempted 'single' (not multiple) Pv neuronal stimulation by focal light stimulation of the cell body (473 nm, 5 ms, 10 Hz) using a pattern-illumination system and recorded firing responses of the stimulated and neighboring Pv neurons in *Shank2⁻/⁻* mPFC slices (prelimbic, layer 2/3).

Intriguingly, burst firings, defined by clustered firings at frequency >80 Hz, were increased by ~50% in stimulated

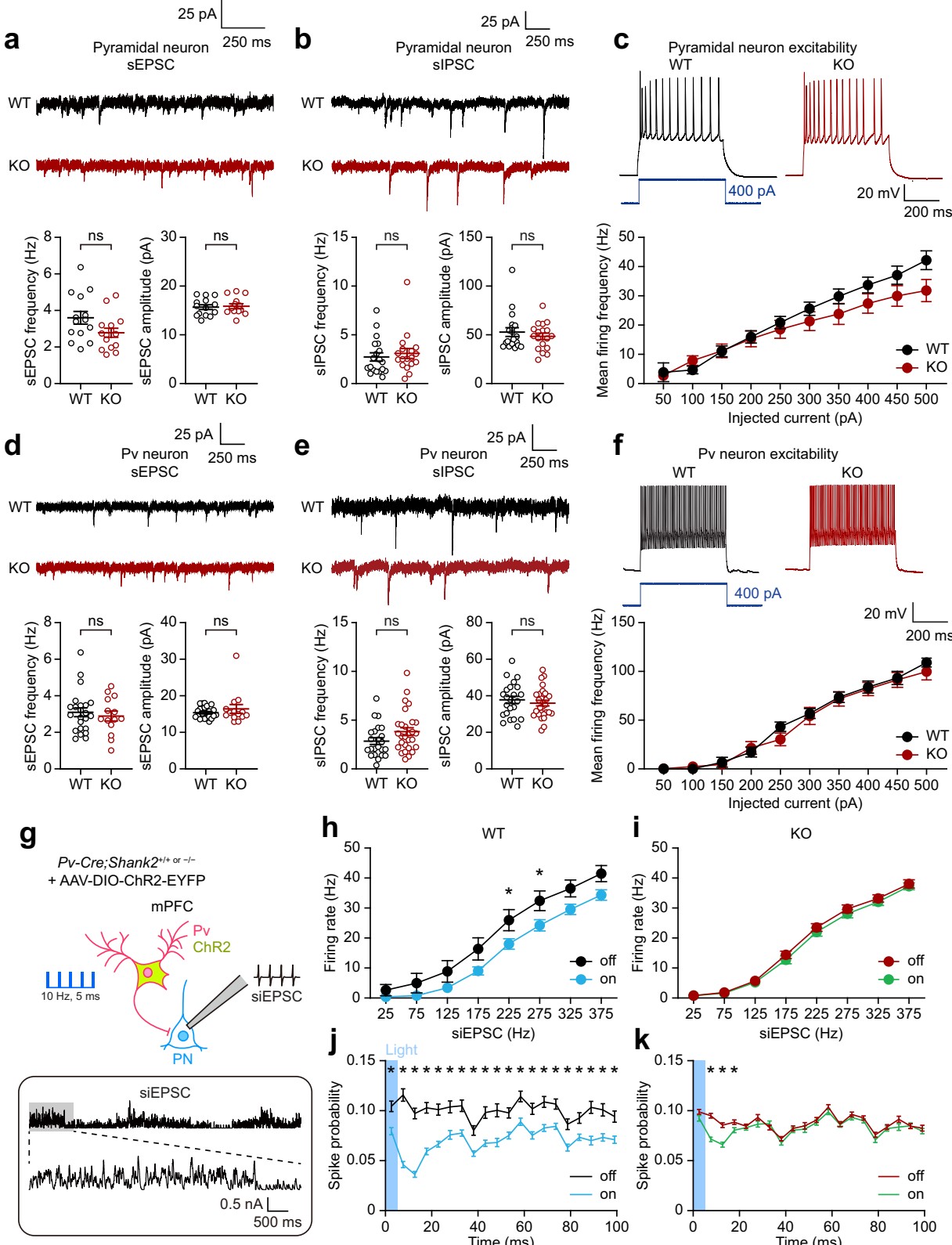

Shank2[−/−] Pv neurons, compared with those in WT Pv neurons, whereas tonic firings were similar, and no firings were ~50% weaker (Fig. 5a). These changes did not lead to an increase in the mean-firing rate in Shank2[−/−] Pv neurons (Fig. 5a).

When firings were measured in neighboring (indirectly stimulated) Shank2[−/−] Pv neurons, burst firings were strongly increased (~8.0-fold) whereas tonic firings were similar, and no

firings were moderately (~25%) decreased, compared with WT Pv neurons (Fig. 5b). These changes accompanied a significant increase (~2.6-fold) in the mean-firing rate. These differences in burst firing and firing rate in WT and Shank2[−/−] Pv neurons did not involve altered intrinsic properties, or different photocurrents, in Pv neurons (Supplementary Table 2, Supplementary Fig. 6a) or altered synaptic activity, as shown by the moderate

**Fig. 4 Stimulated *Shank2*<sup>−/−</sup> Pv neurons fail to inhibit target PNs while displaying unaltered baseline synaptic transmission and excitability. a** Normal frequency and amplitude of sEPSCs in pyramidal neurons (PNs) in the *Shank2*<sup>−/−</sup> mPFC (prelimbic, layer 2/3, 12–13 weeks) ($n = 14$ neurons, 4 mice [WT], 15/4 [KO], ns, not significant, two-tailed Student's *t*-test [frequency], two-tailed Mann–Whitney test [amplitude]). **b** Normal sIPSCs in *Shank2*<sup>−/−</sup> mPFC PNs ($n = 19/6$ [WT], 18/4 [KO], ns, not significant, two-tailed Mann–Whitney test). **c** Normal excitability in *Shank2*<sup>−/−</sup> mPFC PNs, as indicated by input-output curve ($n = 21/5$ [WT], 13/5 [KO], ns, not significant, two-way RM-ANOVA). **d** Normal sEPSCs in *Shank2*<sup>−/−</sup> mPFC Pv neurons ($n = 13/4$ mice [WT], 15/4 [KO], ns, not significant, two-tailed Student's *t*-test [frequency], two-tailed Mann–Whitney test [amplitude]). **e** Normal sIPSCs in *Shank2*<sup>−/−</sup> mPFC Pv neurons. Pv neurons were visualized by 3-week AAV-DIO-ChR2-EYFP infection ($n = 23/6$ [WT], 28/8 [KO], ns, not significant, two-tailed Student's *t*-test). **f** Normal excitability in *Shank2*<sup>−/−</sup> mPFC Pv neurons ($n = 24/6$ [WT], 19/4 [KO], two-way RM-ANOVA). **g–i** WT Pv neuronal stimulation at 10 Hz (ChR2, 473 nm, 5 ms) inhibits PN firings in the mPFC (prelimbic, layer 2/3), whereas *Shank2*<sup>−/−</sup> Pv neuronal stimulation fails to inhibit PN firings, as shown by firing rates plotted across simulated EPSCs (siEPSCs), involving random current injections similar to in vivo situations[21]. mPFC Pv neurons infected with AAV-DIO-ChR2-EYFP were stimulated with siEPSC injections in the presence or absence of Pv neuronal light stimulation ($n = 8/3$ [WT-off], 9/3 [WT-on], 17/6 [KO-off] and 17/6 [KO-on], *$p < 0.05$, two-way ANOVA with Tukey's test and Mann–Whitney test). **j, k** Pv neuronal stimulation suppresses PN discharges in both WT and *Shank2*<sup>−/−</sup> mPFCs during the short (~100 ms) time window immediately after 5-ms light stimulation, although the changes were smaller and normalized within ~20 ms after stimulation in *Shank2*<sup>−/−</sup> Pv neurons ($n = 8/3$ [WT-off], 9/3 [WT-on], 17/6 [KO-off], and 17/6 [KO-on], *$p < 0.05$, two-tailed Mann–Whitney test). See Source data for raw data values and Supplementary Table 1 for statistical details.

impacts of all synaptic blockers (excitatory and inhibitory) on burst firing in *Shank2*<sup>−/−</sup> Pv neurons (Supplementary Fig. 6b). Therefore, *Shank2*<sup>−/−</sup> Pv neurons tend to generate stronger bursts upon light stimulation, compared with WT Pv neurons.

These in vitro results were further supported by in vivo opto-tagging experiments. Pv neuronal stimulation (473 nm, 5 ms, 10 Hz) in the mPFC of freely behaving *Shank2*<sup>−/−</sup> mice induced strong burst firing in stimulated Pv neurons (Supplementary Fig. 7). Specifically, *Shank2*<sup>−/−</sup> Pv neurons displayed repetitive firing that continued to occur after the 5-ms light-stimulation epochs during target encounters (S/O/E), unlike WT Pv neurons, which mainly displayed light-entrained tonic firings.

Pv neuronal stimulation at 40 Hz induces neuronal and glial responses that attenuate Alzheimer's disease-related pathophysiology[68]. We thus tested if 40-Hz stimulation also evokes burst firing in *Shank2*<sup>−/−</sup> Pv neurons, but found that the 40-Hz stimulation was not as efficient as 10-Hz stimulation, without changing burst firing and mean-firing rate in stimulated Pv neurons and increasing tonic, but not burst, firing and mean-firing rate in neighboring Pv neurons (Fig. 5c, d). In an additional experiment comparing 10-, 40-, and 80-Hz stimulations, 80-Hz stimulation minimally induced burst or tonic firing in WT or *Shank2*<sup>−/−</sup> Pv neurons (Supplementary Fig. 8).

**Gap junctional hyperactivity in *Shank2*<sup>−/−</sup> Pv neurons.** The enhanced burst firing in stimulated and neighboring *Shank2*<sup>−/−</sup> Pv neurons might involve gap junctions, known to mediate electrical coupling between Pv neurons to modulate neuronal firing and network activity[31–34,69].

Our computational model simulations revealed that the number and strength of gap junctions in a Pv neuron could be positively correlated with the burst-firing ratio and mean-firing rate in stimulated and neighboring Pv neurons (Fig. 6a; Supplementary Fig. 9a, b). Lowering the total number of Pv neurons in the network with unaltered number and strength of gap junctions increased burst firing (Fig. 6a; Supplementary Fig. 9c, d), suggesting that Pv neuronal density is a contributing factor, although it was not significantly changed in *Shank2*<sup>−/−</sup> mice (Supplementary Fig. 5). Moreover, 40-Hz stimulation was much less efficient than 10-Hz stimulation in evoking bursts in stimulated and neighboring Pv neurons, likely attributable to the low-pass-filtering characteristics of gap junctions.

To directly test if gap junctions are involved in the burst hyperactivity in *Shank2*<sup>−/−</sup> Pv neurons, we employed mefloquine, an inhibitor of the gap junction protein connexin-36 that is abundantly expressed in the brain[70]. Mefloquine decreased burst firing, tonic firing, and mean-firing rate in *Shank2*<sup>−/−</sup> Pv neurons

neighboring a light-stimulated Pv neuron in mPFC slices (Fig. 6b). In WT Pv neurons, mefloquine minimally affected burst firing, although it decreased tonic firing and mean-firing rate. This result from WT neurons is likely attributable to the low baseline gap junctional activity because mefloquine does decease burst firing in WT Pv neurons when baseline gap junctional activity is elevated first (see below; Fig. 7d). The difference in gap junctional activity in WT and *Shank2*<sup>−/−</sup> Pv neurons does not seem to involve altered dendritic arborization because there was no genotype difference (Supplementary Fig. 10). These results suggest that gap junctional hyperactivity promotes burst firing in *Shank2*<sup>−/−</sup> Pv neurons.

**NMDARs cooperate with gap junctions to promote Pv neuronal burst firing.** *Shank2*<sup>−/−</sup> mice show decreased NMDAR activity in principal excitatory neurons in the cortex and hippocampus that are causally associated with social deficits[15] and early postnatal NMDAR hyperactivity[71]. However, recent studies have shown that *Shank2* expression in GABA neurons is likely to be more important than glutamate neuronal *Shank2* expression for social functions[72], although related GABA neuron types and the underlying mechanisms remain unclear.

We thus tested if *Shank2*<sup>−/−</sup> Pv neurons displayed NMDAR dysfunction by patch-clamp recordings. The ratio of NMDAR- and AMPA receptor (AMPAR)-mediated evoked EPSCs (NMDA/AMPA ratio) was markedly decreased (Fig. 7a). This result, together with the normal levels of AMPAR-mediated mEPSCs in Pv neurons (Fig. 7b), suggests a selective decrease in NMDAR function in *Shank2*<sup>−/−</sup> Pv neurons.

We next tested if the NMDAR dysfunction in *Shank2*<sup>−/−</sup> Pv neurons were associated with their impaired burst firing by treating the neurons in slices with the NMDAR agonist D-cycloserine, which rescues social deficits in *Shank2*<sup>−/−</sup> mice[15]. D-cycloserine (DCS) treatment substantially increased burst firing and mean-firing rate in *Shank2*<sup>−/−</sup> Pv neurons neighboring the stimulated Pv neuron (Fig. 7c). In WT Pv neurons, DCS treatment induced a similar increase in burst firing in neighboring Pv neurons. In addition, the DCS-dependent burst enhancement was reversed by blocking gap junctional activity with mefloquine in both WT and *Shank2*<sup>−/−</sup> Pv neurons (Fig. 7d). Intriguingly, directly simulated *Shank2*<sup>−/−</sup> Pv neurons did not show increased burst firing upon D-cycloserine treatment (Supplementary Fig. 11). These results suggest that gap junctions, linking the two (stimulate and neighboring) Pv neurons, may act in concert with excitatory synapses, likely on neighboring neuronal dendrites, to promote burst firing in neighboring neurons.

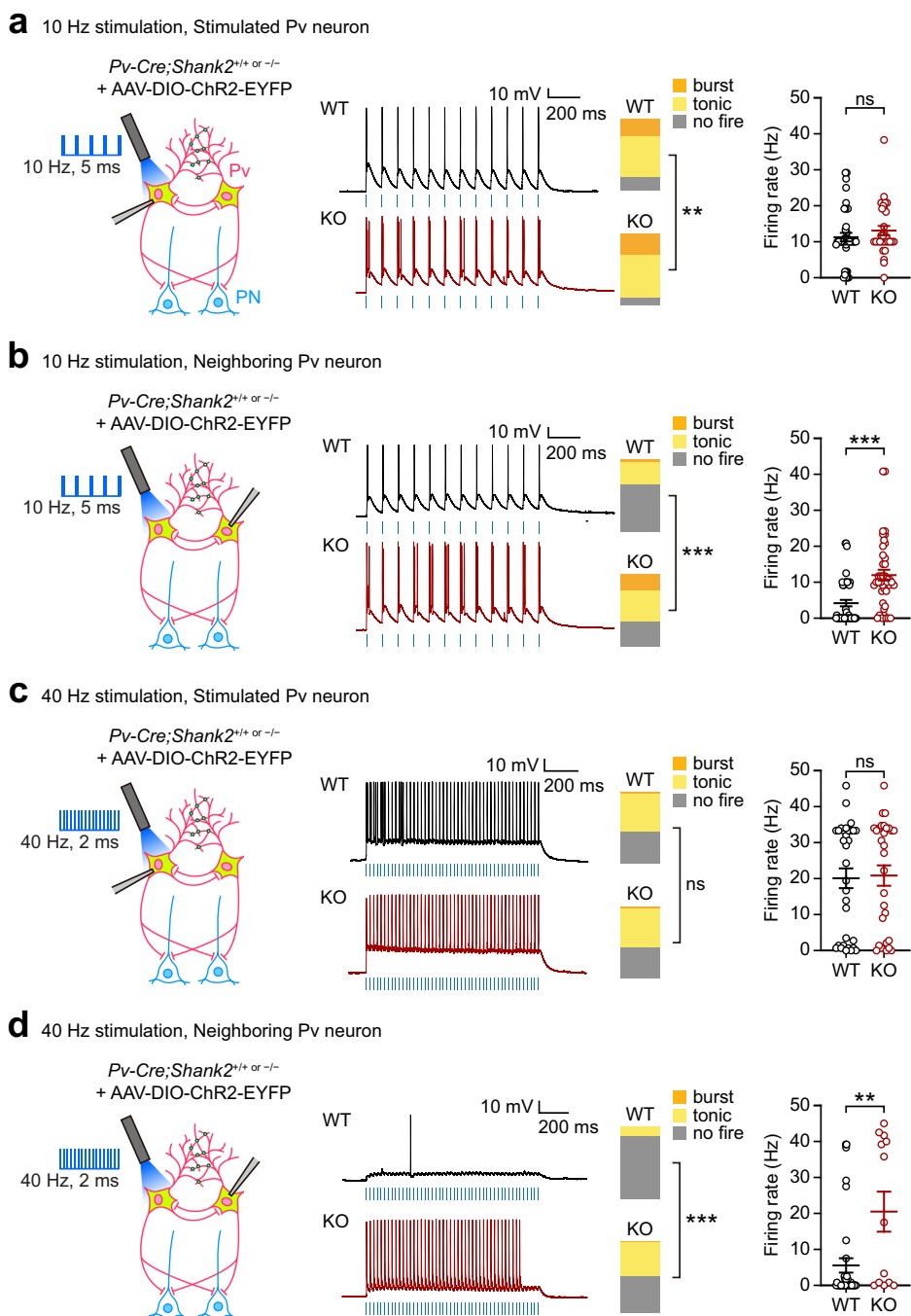

**Fig. 5 Burst firing in *Shank2⁻/⁻* Pv neurons strongly induced by 10-Hz, but not 40-Hz, stimulation. a** Increased burst firings in single Pv neurons focally light-stimulated in the cell body in the *Shank2⁻/⁻* mPFC (prelimbic, layer 2/3), as shown by proportions of neurons with specific firing patterns (no-fire, tonic-fire, burst-fire). Note that the mean-firing rate shows only an increasing tendency. A single Pv neuron in acute slices from Pv-Cre;*Shank2⁻/⁻* or Pv-Cre;WT (control) mice (>12 weeks) infected in the mPFC with AAV-DIO-ChR2-EYFP (3 weeks) was stimulated by blue light (473 nm, 10 Hz, 5 ms, 12 epochs) ($n = 42$ neurons from 9 mice [WT], 31/12 [KO], **$p < 0.01$, ns, not significant, Chi-square test [proportion], two-tailed Mann–Whitney test [firing rate]; $p = 0.0028$ [proportion], $p = 0.2561$ [firing rate]). **b** Increased burst firings in Pv neurons neighboring a light-stimulated single Pv neuron in the *Shank2⁻/⁻* mPFC, as shown by proportions of firing patterns and mean-firing rates ($n = 47/10$ [WT], 42/15 [KO], ***$p < 0.001$, Chi-square test [proportion], two-tailed Mann–Whitney test [firing rate]; $p < 0.0001$ [proportion], $p < 0.0001$ [firing rate]). **c, d** Limited burst firings in light-stimulated and neighboring Pv neurons by 40 Hz stimulation in the *Shank2⁻/⁻* mPFC ($n = 32/8$ [WT], 30/10 [KO] for light-stimulated Pv neurons, 37/9 [WT], 17/9 [KO] for neighboring Pv neurons, **$p < 0.01$, ***$p < 0.001$, ns, not significant, Chi-square test [proportion], two-tailed Mann–Whitney test [firing rate]; $p = 0.7524$ [proportion-c], $p = 0.671$ [firing rate-c], $p < 0.0001$ [proportion-d], $p = 0.0067$ [firing rate-d]). See Source data for raw data values and Supplementary Table 1 for statistical details.

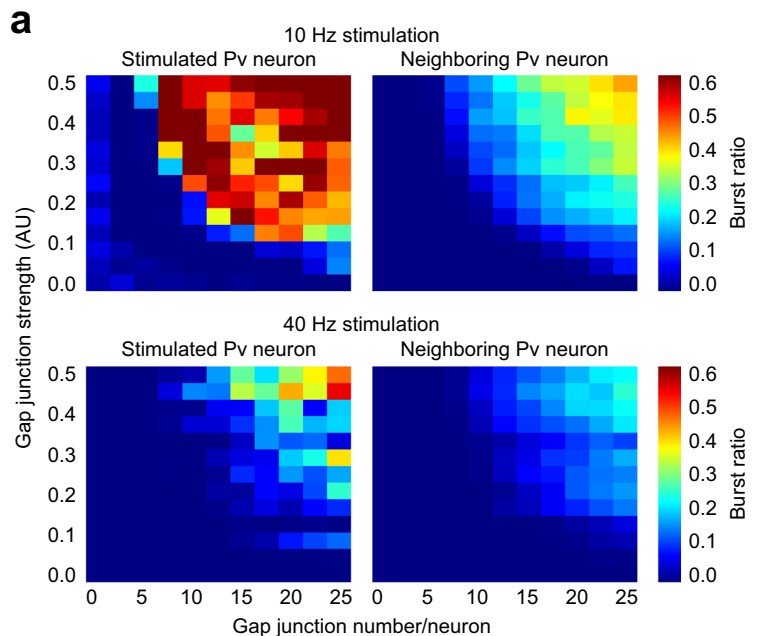

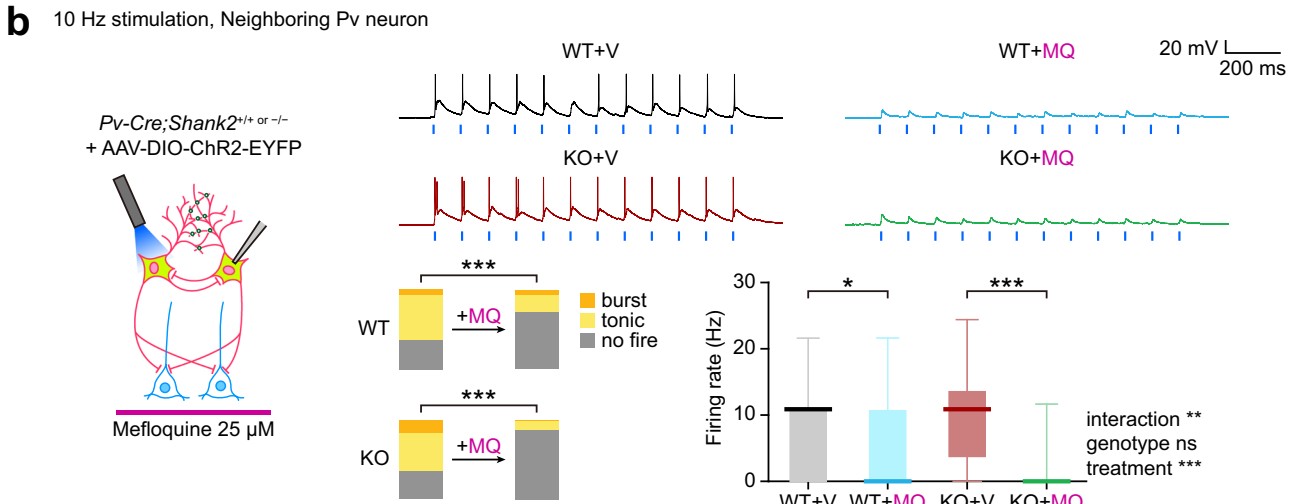

**Fig. 6 Gap junction-dependent burst firing in _Shank2⁻/⁻_ Pv neurons. a** Computational model simulations demonstrate that the number and the strength of individual gap junctions of a Pv neuron are positively correlated with the extents of Pv neuronal burst firings, as monitored in light-stimulated single Pv neurons or in Pv neurons neighboring a light-stimulated single Pv neuron. Note that 40-Hz stimulation is less effective than 10-Hz stimulation for inducing burst firings, probably due to the low-pass filtering characteristics of gap junctions. **b** Mefloquine (25 μM) treatment decreases burst and tonic firings and firing rates in _Shank2⁻/⁻_ Pv neurons neighboring a light-stimulated single Pv neuron, whereas it minimally affects burst firing in WT Pv neurons, although tonic firing and firing rate are decreased (prelimbic, layer 2/3, 12–13 weeks). Data: minimal, maximal, median, 25%, and 75% values (_n_ = 40 neurons from 13 mice [KO-V/vehicle, before mefloquine/MQ treatment; KO-MQ/mefloquine, after MQ treatment] and 29 neurons from 13 mice [WT-V and WT-MQ], *$p < 0.05$, ***$p < 0.001$, ns, not significant, Chi-square test [proportion], two-way ANOVA with Sidak's test [firing rate]; $p < 0.0001$ [WT-proportion], $p < 0.0001$ [KO-proportion], interaction $p$ 0.0051, drug $p < 0.0001$, group $p$ 0.8354 [firing rate], WT $p = 0.0105$, KO $p < 0.0001$ [Veh-Mef, firing rate]). See Source data for raw data values and Supplementary Table 1 for statistical details.

These results collectively suggest that NMDARs act in concert with gap junctions to promote Pv neuronal burst firing, as supported by the differences between directly stimulated and neighboring Pv neurons. In addition, these results suggest that the decreased NMDAR activity in _Shank2⁻/⁻_ Pv neurons leads to the inhibition of burst firing and the D-cycloserine-sensitive social deficits in _Shank2⁻/⁻_ mice.

**Boosting Pv neuronal burst improves social cognition and behavior.** Light-stimulated _Shank2⁻/⁻_ Pv neurons fail to inhibit target PNs in slice preparations (Fig. 4g–k), but these neurons

display increased burst firing that requires gap junctions and is facilitated by NMDARs (Figs. 6 and 7). To test if the increased burst firing has any beneficial effects on social cognition and behavior, we attempted boosting burst firing in _Shank2⁻/⁻_ Pv neurons by 10-Hz light stimulation.

To this end, _Pv-Cre;Shank2⁻/⁻_ mice were injected with AAV-Dio-ChR2-EYFP in the mPFC and light-stimulated at 10 Hz (473 nm, 5 ms) during social interaction in the conventional three-chamber test[73] (Fig. 8a). This stimulus restored social interaction in _Shank2⁻/⁻_ mice (Fig. 8b, c; Supplementary Fig. 12) without affecting locomotor activity or anxiety-like behavior (Supplementary Fig. 13a–d).

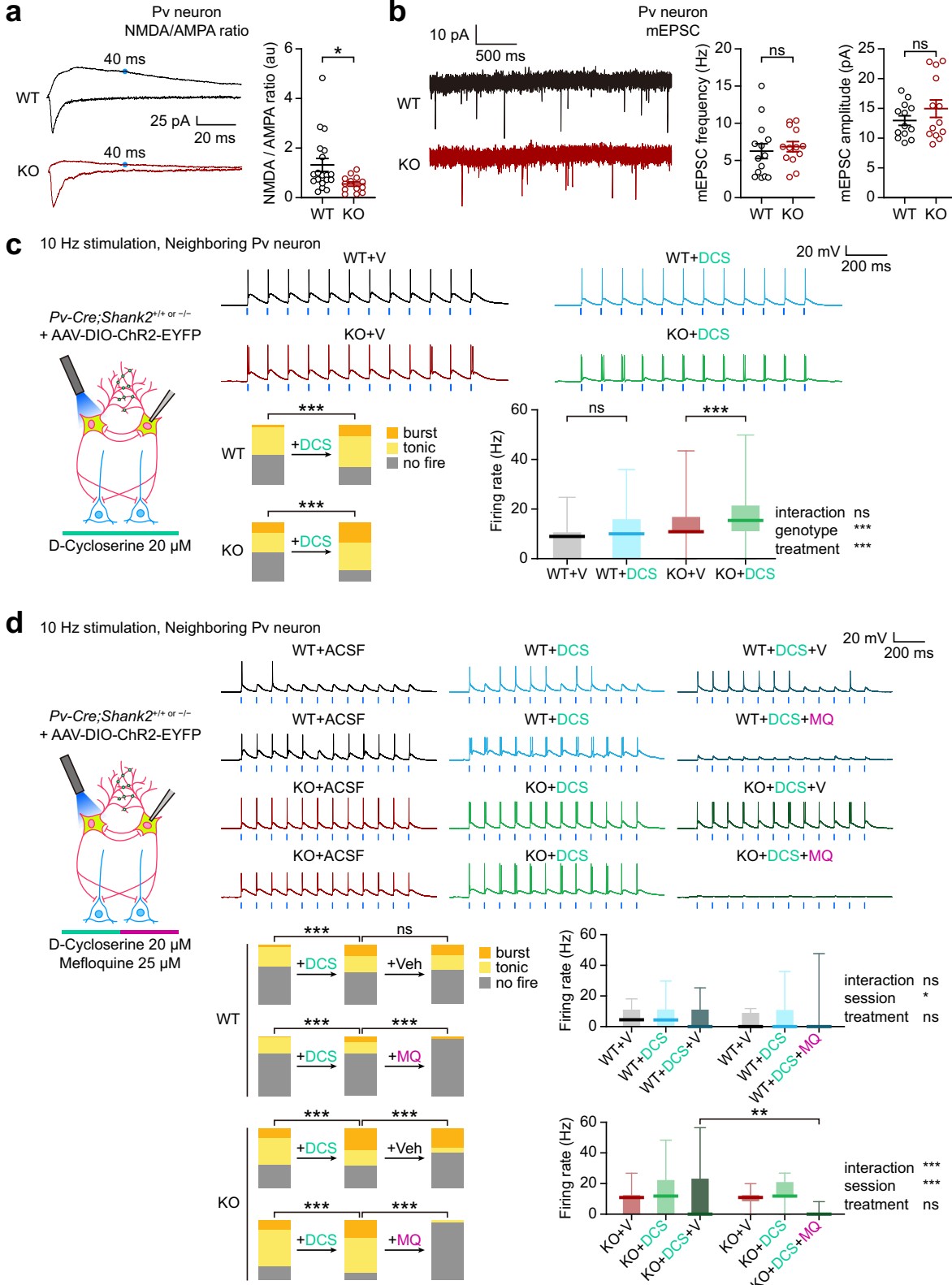

In contrast, Pv neuronal stimulation at 40 Hz, an inefficient inducer of bursts in *Shank2*[−/−] Pv neurons, using ChETA (channelrhodopsin-2 containing an E123T mutation) failed to rescue social interaction in *Shank2*[−/−] mice, despite that it could drive Pv neuronal firings at 40 Hz in vivo (Fig. 8d; Supplementary Fig. 14a, b). Using ChR2, instead of ChETA, to drive 40-Hz Pv neuronal firings also failed to rescue social deficits in *Shank2*[−/−]

mice (Supplementary Fig. 13e). In addition, continuous Pv neuronal stimulation using SSFO (stable step-function opsin) did not rescue social interaction, despite that it could suppress target neuronal firings (Fig. 8e; Supplementary Fig. 14c), suggesting that continuous stimulation does not efficiently evoke burst firing in Pv neurons.

In addition, using single-unit recordings, we tested whether 10-Hz Pv neuronal stimulation also rescued cortical

**Fig. 7 NMDARs cooperate with gap junctions to promote Pv neuronal burst firing. a** Decreased ratio of NMDAR- and AMPAR-mediated evoked EPSCs in *Shank2*$^{-/-}$ Pv neurons, compared with WT Pv neurons (prelimbic, layer 2/3, 12–13 weeks) ($n = 19$ neurons from 3 mice [WT] and 13/2 [KO], *$p < 0.05$, two-tailed Mann–Whitney test; $p = 0.0125$ [NMDA/AMPA ratio]). **b** Normal levels of mEPSCs in *Shank2*$^{-/-}$ Pv neurons, compared with WT Pv neurons (prelimbic, layer 2/3, 12–13 weeks) ($n = 14/4$ [WT] and 13/4 mice [KO], ns, not significant, two-tailed Student's *t*-test). **c** D-cycloserine (DCS; 20 μM) induces increases in burst firing in WT and *Shank2*$^{-/-}$ Pv neurons neighboring a light-stimulated single Pv neuron (prelimbic, layer 2/3, 12–13 weeks), as shown by proportions of burst/tonic/no-firing neurons and mean-firing rate. Data: minimal, maximal, median, 25%, and 75% values ($n = 43/18$ [WT/V followed by WT/DCS] and 38/19 [KO/V followed by KO/DCS], ***$p < 0.001$, ns, not significant, Chi-square test [proportion], two-way ANOVA with Sidak's test [firing rate]; $p < 0.0001$ [WT-proportion], $p < 0.0001$ [KO-proportion], interaction $p = 0.0653$, drug $p < 0.0001$, group $p = 0.0003$ [firing rate], WT $p = 0.1654$, KO $p < 0.0001$ [Veh-DCS, firing rate]). **d** Mefloquine reverses DCS-dependent enhancement of burst firing in WT and *Shank2*$^{-/-}$ Pv neurons neighboring a light-stimulated single Pv neuron (prelimbic, layer 2/3, 12–13 weeks), as shown by proportions of burst/tonic/no-firing neurons, although mean-firing rates were not changed. WT and *Shank2*$^{-/-}$ Pv neurons were recorded of firings sequentially in the presence of vehicle and DCS, followed by mefloquine (25 μM) or vehicle treatment. Data: minimal, maximal, median, 25%, and 75% values ($n = 16/10$ [WT-DCS-V], 27/10 [WT-DCS-MQ], 21/12 [KO-DCS-V], and 20/12 [KO-DCS-V], **$p < 0.01$, ***$p < 0.001$, Chi-square test [proportion], two-way ANOVA with Sidak/Tukey test [firing rate; see Supplementary Table 1 for details]; WT-DCS $p < 0.0001$, DCS-Veh $p = 0.8077$ [WT+DCS+Veh-proportion], WT-DCS $p < 0.0001$, DCS-Mef $p < 0.0001$ [WT+DCS+Mef-proportion], KO-DCS $p < 0.0001$, DCS-Veh $p < 0.0001$ [KO+DCS+Veh-proportion], KO-DCS $p = 0.0001$, DCS-Mef $p < 0.0001$ [KO+DCS+Mef-proportion], interaction $p = 0.3942$, session $p = 0.0430$, treatment $p = 0.1332$, Veh-Mef $p = 0.2346$ [WT-firing rate], interaction $p < 0.0001$, session $p < 0.0001$, treatment $p = 0.1626$, Veh-Mef $p = 0.0086$ [KO-firing rate]). See Source data for raw data values and Supplementary Table 1 for statistical details.

representation of social/non-social contexts (Fig. 8f, g; Supplementary Fig. 15a). This analysis indicated that 10-Hz Pv neuronal stimulation modestly decreased the proportion of mPFC neurons specific for O (but not for S, E, or Si) targets, and decreased the total number of targets (social and non-social) that a single neuron can discriminate in *Shank2*$^{-/-}$ mice (Fig. 8h, i; Supplementary Fig. 15b). In contrast, WT mPFC neurons were largely unaffected by the same stimulation, i.e., unaltered total number of targets to which a single neuron can respond, although responses to sideness and social targets were increased (Fig. 8h; Supplementary Fig. 15c).

The baseline differences in measures of discriminative/correlative firings between EYFP-expressing WT and *Shank2*$^{-/-}$ mPFC neurons described above (Fig. 8f–i) partly differ from the results from naïve animals (Fig. 1d, e). This could be attributable to the different genetic backgrounds of *Shank2*$^{-/-}$ and *Pv-Cre;Shank2*$^{-/-}$ mice (C57BL6/N and C57BL6/N + C57BL6/J, respectively) and AAV-mediated gene delivery in *Pv-Cre;Shank2*$^{-/-}$ mice but not in *Shank2*$^{-/-}$ mice. Together, these results collectively suggest that boosting Pv neuronal burst by 10-Hz light stimulation improves cortical social representation and behavioral social interaction.

**Pv neuronal bursts suppress cortical spike-wave synchrony in the *Shank2*$^{-/-}$ mPFC.** The results described thus far suggest the possibility that the increased gap junctional activity in *Shank2*$^{-/-}$ Pv neurons might represent a compensatory effort to produce strong burst firing upon Pv neuronal stimulation during social interaction, which, however, failed to rescue social interaction under physiological conditions although it could do so in the presence of optogenetic 10-Hz Pv neuronal stimulation. Alternatively, the increased gap junctional activity might function as a pathological mechanism that is actively engaged in disrupting mPFC neuronal activity and social cognition, resulting in impaired social interaction. To differentiate between these two possibilities, we measured the extent of the synchrony between the timing of neuronal firings and particular phases in the waveforms of the local field potentials, known as the spike-wave synchrony and thought to regulate brain functions, including attention[55], cognitive flexibility[74], sensory processing[63], and social novelty recognition[26].

Specifically, we compared the spike-wave synchrony patterns in the mPFC of WT and *Shank2*$^{-/-}$ mice with Pv neurons expressing EYFP (control) or ChR2 in the presence of 10-Hz stimulation. Intriguingly, 10-Hz Pv neuronal stimulation

decreased the spike-wave synchrony in *Shank2*$^{-/-}$ mPFC neurons in frequency ranges including gamma (low and high), theta, and alpha during social interaction (Fig. 9a, b). In contrast, WT mPFC neurons showed only moderate increases in the spike-wave synchrony in the alpha and low-gamma but not other ranges (Fig. 9a, b). These results suggest that Pv neuronal bursts suppress spike-wave synchrony in the *Shank2*$^{-/-}$ mPFC.

**Gap junctional inhibition in mPFC Pv neurons moderately enhances social interaction.** Last, to test the possibility that an increase in the gap junctional activity has any influences on social interaction, we directly infused mefloquine to the mPFC of WT and *Shank2*$^{-/-}$ mice through cannula and found that it does not affect social interaction in WT or *Shank2*$^{-/-}$ mice, as shown by sniffing time and social preferential index (Fig. 9c). This result, however, is difficult to interpret because gap junctions are detected in various GABAergic cell types, including fast-spiking Pv neurons, somatostatin-positive neurons, and multipolar burst cells (a subtype of Pv neurons distinct from fast-spiking Pv neurons)[24,46].

We thus attempted Pv neuron-specific knockdown of connexin-36 expression and gap junctional inhibition in the mPFC of WT mice by injecting two independent AAVs carrying connexin-36 knockdown shRNAs under the S5E2 promoter for selective Pv neuronal gene expression[75]. These two groups of mice similarly showed moderate increases in social interaction in the three-chamber test, as shown by social sniffing and preference index, with normal locomotion or anxiety-like behaviors (Fig. 9d; Supplementary Fig. 16). These results suggest that gap junctional activity in Pv neurons by itself has negative influences on social interaction.

## Discussion
One of the key conclusions from our study is that NMDAR hypofunction impairs cortical Pv neuronal functions, specifically, burst firing and local oscillation, leading to abnormal cortical social representation and behavioral social interaction.

Previous studies on animal models of ASD have demonstrated that limited NMDAR function can cause autistic-like behaviors[11–16]. In addition, impaired GABA neuronal functions have been associated with autistic-like behaviors in a number of animal models of ASD[1,17–23]. Our study connects these two emerging mechanisms of ASD, suggesting the need to further study NMDAR dysfunctions in

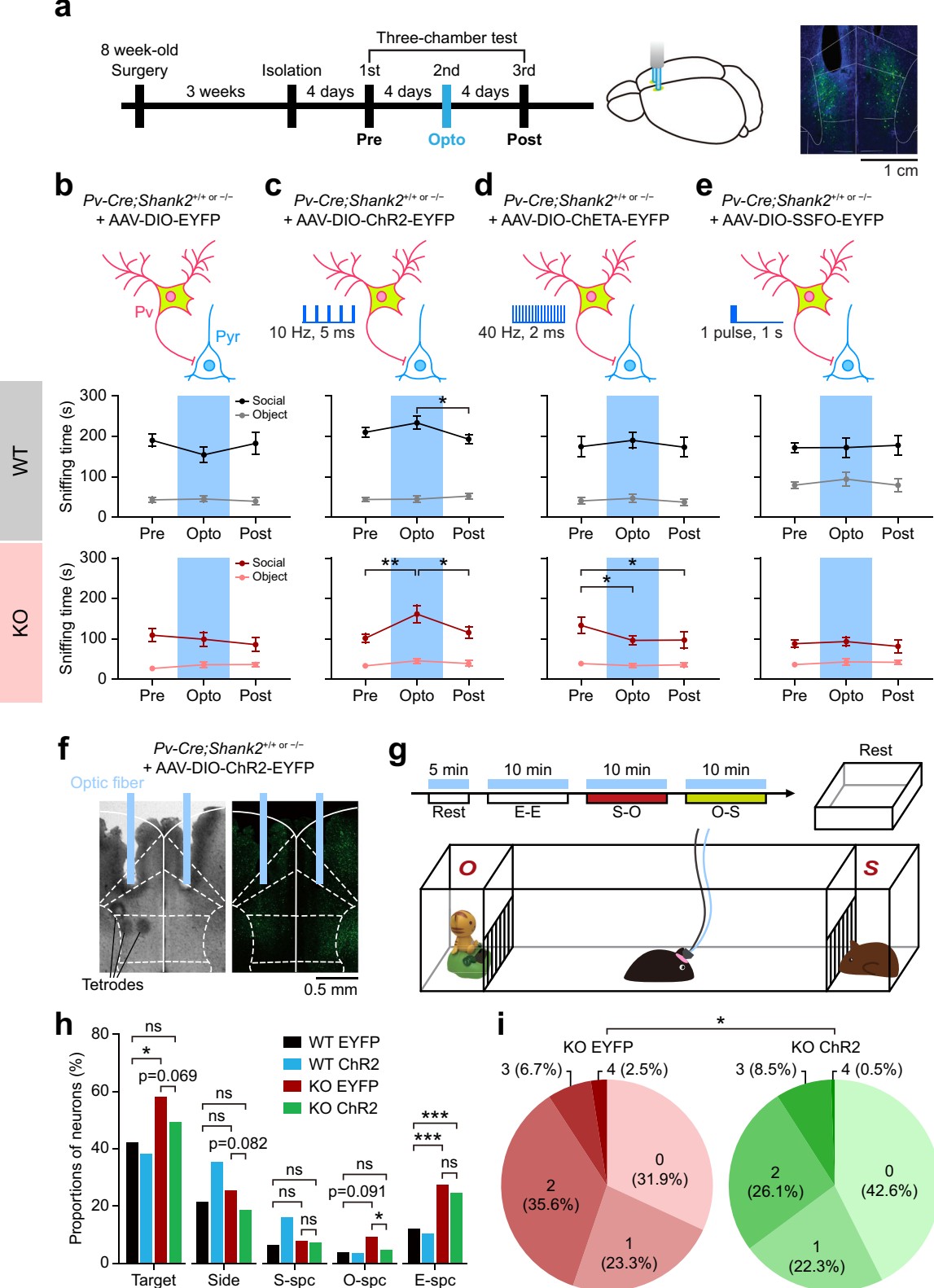

GABA neurons, including Pv neurons, and related mechanisms in future studies.

In addition to identifying NMDAR hypofunction in $Shank2^{-/-}$ Pv neurons, our results indicate how NMDAR hypofunction impairs Pv neuronal functions by demonstrating suppressed burst firing. Burst firing is a functional aspect of Pv neurons that has been understudied, as compared with other Pv neuronal

functions such as synaptic inhibition of target PNs and Pv-to-Pv cross-inhibition, known to cooperatively shape local network activity[21,25,59–63]. Specifically, our results indicate that burst-firing activity is decreased in Pv neurons in the mPFC of behaving $Shank2^{-/-}$ mice engaged in social interaction, whereas baseline Pv neuronal functions such as spontaneous inhibitory synaptic transmissions are normal. Burst firing promotes the reliability of

**Fig. 8 Burst-evoking Pv neuronal stimulation improves social interaction and cortical social representation. a** Experimental scheme for social rescue by Pv neuronal stimulation. *Pv-Cre;WT* and *Pv-Cre;Shank2⁻/⁻* mice (8 weeks) were bilaterally injected in the mPFC with AAV-DIO-ChR2/ChETA/SSFO-EYFP or AAV-DIO-EYFP (control) for 3 weeks prior to three-chamber experiments. Experiments consisted of three rounds of three-chamber tests in which the second (Opto) round, but not the first (Pre) or third (Post) round, was performed in the presence of blue-light stimulation throughout the 10-min S–O session. Right, examples of mPFC neurons from a *Pv-Cre;WT* mouse injected with AAV-DIO-EYFP; DAPI (blue), nuclear marker. **b–e** Pv neuronal stimulation in the mPFC at 10 Hz (5 ms; ChR2), but not in a continuous mode (SSFO) or at 40 Hz (2 ms; ChETA), improves three-chamber social interaction in Pv-Cre;*Shank2⁻/⁻* mice infected in the mPFC with the indicated viruses (>12 weeks + 3 weeks), as shown by time spent sniffing (or nose-poking) social and object targets ($n = 15$ mice [WT-EYFP], 11 [KO-EYFP], 20 [WT-ChR2-10-Hz], 16 [KO-ChR2-10-Hz], 8 [WT-ChETA], 10 [KO-ChETA], 15 [WT-SSFO], 14 [KO-SSFO], *$p < 0.05$, **$p < 0.01$, two-way RM-ANOVA with Tukey's test). **f–i** 10-Hz Pv neuronal stimulation modestly improves neuronal representation of social and non-social contexts in the mPFC of *Shank2⁻/⁻* mice, as shown by the proportions of neurons that show discriminative firings to target pairs (S–O, S–E, E–O, and L–R/left–right) and targets (S, E, O, Si, and total) and the total number of target pairs that a single neuron can discriminate (S–O, S–E, E–O, and L–R; ranging from 0 to 4). Two-way ANOVA could not be performed because the neuronal and target-number proportions were extracted from the calculation of target pair-discriminative or target-selective neurons using Venn diagrams; we thus attempted Chi-square analysis with Bonferroni correction and multiple comparisons with *p*-value adjustments ($n = 194$ neurons from 7 mice for WT-EYFP, 72/8 [WT-ChR2], 163/4 [KO-EYFP], and 364/9 [KO-ChR2], *$p < 0.017$ [comparison of WT-EYFP, KO-EYFP, KO-ChR2], Chi-square test with Bonferroni correction). See Source data for raw data values and Supplementary Table 1 for statistical details.

information transfer between neurons[35–41], suggesting that the decreased burst firing in *Shank2⁻/⁻* Pv neurons may impair their output functions such as Pv-to-PN inhibition and Pv-to-Pv cross-inhibition during behavioral social interaction that fine-regulate PN firings and local oscillations. In support of this possibility, the 10-Hz (but not 40-Hz) Pv neuronal stimulation, which induces strong burst firing, rescues social interaction in *Shank2⁻/⁻* mice.

If the light-stimulated *Shank2⁻/⁻* Pv neurons promoted the Pv-to-PN inhibition in the mPFC it could have increased Pv neuronal 'burst' firing to control the timing of pExc neuronal firing[76] induced by social stimuli in a manner ensuring reliable information transfer onto target pyramidal neurons of Pv neurons[36,37]. However, the Pv-to-PN inhibitory pathway was still substantially impaired in *Shank2⁻/⁻* mice even in the presence of strong light-dependent Pv neuronal stimulation (although in slice preparations), a stimulation that could rescue social cognition and interaction in *Shank2⁻/⁻* mice. Therefore, Pv-to-Pv cross-inhibition, rather than Pv-to-PN inhibition, might be more strongly improved by light-dependent Pv neuronal stimulation. Alternatively, the light stimulation may improve the network activity of Pv neurons such as local oscillations (i.e., gamma rhythms) and spike-wave synchrony (see below) without much involving Pv-dependent synaptic inhibition, as shown by the largely intact burst firing or gap junctional activity in the presence of inhibitory synaptic blockade (Supplementary Fig. 6b).

Importantly, our results suggest gap junctional activity as a key mechanism that acts cooperatively with NMDAR to promote Pv neuronal bursts. This conclusion is supported by the decreased burst firing induced by the gap junction inhibitor mefloquine. In addition, the NMDAR agonist D-cycloserine promotes burst firing more strongly in *Shank2⁻/⁻* Pv neurons that display higher gap junctional activity relative to WT Pv neurons. Moreover, the DCS-dependent burst enhancement is reversed by mefloquine-dependent gap junctional inhibition. Gap junctions are known to regulate electrical coupling between GABA neurons, including Pv neurons, and local network activities[31–34]. Therefore, NMDARs seem to act in concert with gap junctions to regulate Pv neuronal burst firing. It is possible that excitatory drive or signaling events mediated/triggered by NMDARs in Pv neuronal dendrites may act cooperatively with gap junctions to promote burst firing in the downstream axon hillock region.

Notably, our data indicate that gap junctional hyperactivity is associated with NMDAR hypoactivity and suppressed burst firing in *Shank2⁻/⁻* mice. The gap junctional protein connexin-36 is highly expressed at early postnatal stages in the brain but gradually decreases to adult levels[31]. This age-dependent change is

known to require NMDAR activity at least in the hippocampus[77]. Our previous results indicate that NMDAR activity is abnormally high in early postnatal stages (around ~P14), which causes NMDAR hypoactivity in juvenile and adult *Shank2⁻/⁻* mice[15,71]. It is therefore tempting to speculate that (1) Pv neuronal output functions are regulated by both gap junctions and NMDARs, (2) early, immature Pv neuronal functions are more strongly supported by gap junctions relative to NMDARs, whereas late, mature Pv neuronal functions are more strongly supported by NMDARs, (3) the limited NMDAR function at juvenile and young adult stages in *Shank2⁻/⁻* Pv neurons may fail to promote age-dependent gap junctional suppression, and (4) the gap junctional hyperactivity in late-stage *Shank2⁻/⁻* Pv neurons may be caused by NMDAR hypoactivity or through compensatory increases that try to maintain total output functions of Pv neurons, involving NMDAR and gap junctional cooperativity, at constant levels. These chronic procedures may be distinct from the acute NMDAR-dependent activation of gap junctional activity, which, however, may chronically suppress the expression of gap junctional proteins to prevent excessive gap junctional activity.

These potential interplays between NMDARs and gap junctions in Pv neurons could be further associated with other modulators in Pv neurons, including mGluR5 and muscarinic receptors. Our previous study on *Shank2⁻/⁻* mice has shown that activation of metabotropic glutamate receptor 5 by a positive allosteric modulator (CDPPB) rescues NMDAR function and social deficits[15]. In addition, Pv neuronal mGluR5 is critical for Pv neuronal development, network activity, and behaviors[78], suggesting the possibility that mGluR5 positive allosteric modulators may improve Pv neuronal function and social interaction in *Shank2⁻/⁻* mice. In addition, given that muscarinic receptor activation can increase gap junctional coupling in adult Pv-positive multipolar bursting cells in frontal and somatosensory cortices (layer 2/3)[46], muscarinic receptor agonists such as carbachol may modulate Pv neuronal burst firing and social interaction in *Shank2⁻/⁻* mice.

The increased gap junctional activity in *Shank2⁻/⁻* Pv neurons may represent a compensatory change to produce stronger burst during social interaction or a pathophysiological change that by itself suppresses cortical social representation and social behavior. Our results from connexin-36 knockdown restricted to Pv neurons in the mPFC suggest that gap junctional activity moderately suppresses social interaction in WT mice. In addition, the spike-wave synchrony is strongly decreased by 10-Hz Pv neuronal stimulation in *Shank2⁻/⁻* mice. These results suggest the possibility that the increased gap junctional activity in *Shank2⁻/⁻* mice

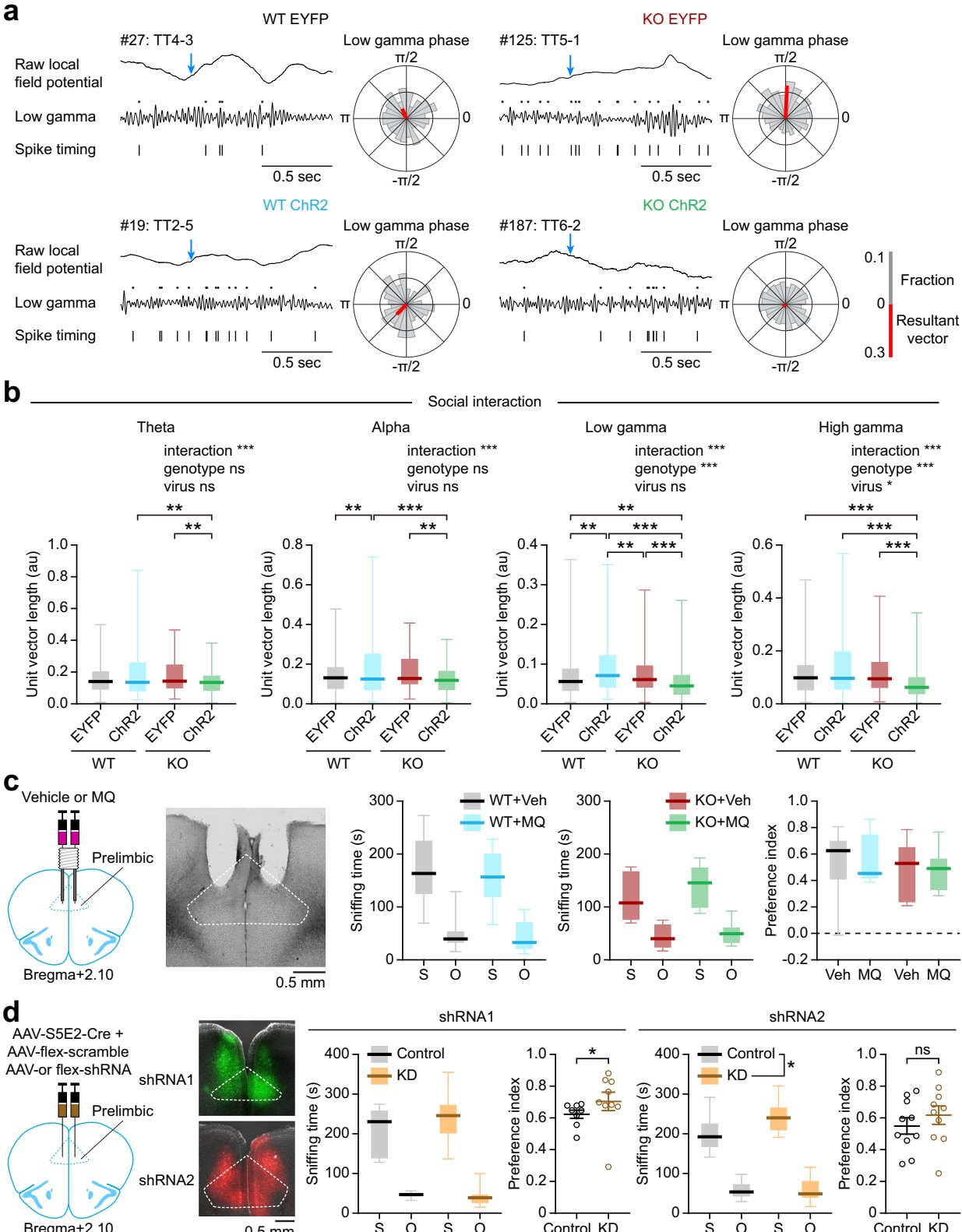

may suppress social interaction by itself through excessive Pv neuronal network synchrony, and that the burst firing induced by 10-Hz Pv neuronal stimulation in *Shank2*[−/−] mice may suppress excessive Pv neuronal network synchrony to promote social interaction.

Last, our study provides the following general conceptual advancements in our understanding of ASD-related neurobiological mechanisms; (1) identification of NMDAR dysfunction in ASD-related mutant Pv neurons, (2) involvement of burst firing in Pv neuronal functions and social cognition, (3) involvement of

**Fig. 9 Pv neuronal bursts suppress cortical spike-wave synchrony in the *Shank2−/−* mPFC, and Pv neuronal gap junctional inhibition moderately enhances social interaction in WT mice. a** Examples of spike-wave synchrony. Low-gamma oscillations extracted from raw LFPs were aligned with individual spikes to determine spike-wave synchrony. Arrows (blue), sniffing (nose-poke) onset for social target; rose plots, polar histograms of phase synchrony of a representative neuron during social interaction; red-colored bars, the sum of individual vectors. **b** Comparisons of spike-wave synchronies for the social target across different frequency ranges in the WT and *Shank2−/−* mPFC in the presence of Pv neuronal stimulation during social interactions. Data: minimal, maximal, median, 25%, and 75% values ($n = 224$ neurons from 7 mice for WT-EYFP, 91/8 [WT-ChR2], 145/4 [KO-EYFP], and 331/9 [KO-ChR2], $**p < 0.01$, $***p < 0.001$, two-way ANOVA with Tukey's test; interaction $p = 0.0003$, genotype $p = 0.17$, virus $p = 0.7$ [theta], interaction $p < 0.0001$, genotype $p = 0.096$, virus $p = 0.939$ [alpha], interaction $p < 0.0001$, genotype $p = 0.096$, virus $p = 0.939$ [low-gamma], interaction $p < 0.0001$, genotype $p = 0.001$, virus $p = 0.025$ [high-gamma]). **c** Direct mefloquine infusion (25 μM) into the mPFC of WT and *Shank2−/−* mice does not affect three-chamber social interaction, as shown by sniffing time for social/object targets and the social preference index ([social-sniffing time − object-sniffing time]/total sniffing time%). Data: minimal, maximal, median, 25%, and 75% values ($n = 11$ mice [T-Veh/vehicle] 11 [WT-MQ/mefloquine], 8 [KO-Veh], and 8 [KO-MQ], two-way RM-ANOVA). **d** Connexin-36 knockdown in Pv neurons in the mPFC of WT mice, using two independent shRNA constructs, moderately increases three-chamber social interaction. AAV-S5E2-Cre + AAV-flex-shRNA1/2-EGFP (or scrambled shRNA control) were injected into the WT mPFC at 14–16 [shRNA1] and 8–10 [shRNA2] weeks, followed by social-interaction tests after 2 weeks. Data: minimal, maximal, median, 25%, and 75% values ($n = 8$ [WT-control_1], 8 [WT-shRNA1], 9 [KO-control_1], and 9 [KO-shRNA1], 10 [WT-control_2], 10 [WT-shRNA2], 10 [KO-control_2], and 10 [KO-shRNA2], $*p < 0.05$, ns, not significant, two-way RM-ANOVA, two-tailed Mann–Whitney test, and two-tailed Student's *t*-test; interaction $p = 0.2335$, treatment $p = 0.3617$, target $p < 0.0001$ [sniffing-shRNA1], $p = 0.0274$ [preference-shRNA1], interaction $p = 0.1258$, treatment $p = 0.0303$, target $p < 0.0001$ [sniffing-shRNA2], $p = 0.3869$ [preference-shRNA2]). See Source data for raw data values and Supplementary Table 1 for statistical details.

gap junctions in Pv neuronal bursts, (4) cooperative actions of NMDARs and gap junctions for Pv neuronal bursts, and (5) identification of gap junctions, additional to NMDARs, as a potential target for ASD therapy.

## Methods
### Key resources table.

| Reagent or resource | Source | Identifier |
|---|---|---|
| *Antibodies* | | |
| Mouse monoclonal anti-parvalbumin | Millipore | MAB1572 |
| Rabbit polyclonal anti-calretinin | Millipore | AB5054 |
| Rabbit ployclonal anti-somatostatin | Peninsula | T-4547 |
| *Bacterial and virus strains* | | |
| AAV2/9-Ef1a-DIO-hChR2(H134R)-EYFP | UNC Vector Core | |
| AAV2/9-Ef1a-DIO-hChR2(C128S/D156A)-EYFP | UNC Vector Core | |
| AAV2/9-Ef1a-DIO-ChETA-EYFP | UNC Vector Core | |
| AAV2/9-Ef1a-DIO-EYFP | UNC Vector Core | |
| *Chemicals, peptides, and recombinant proteins* | | |
| Sodium chloride | Sigma | Cat#S7653 |
| Sodium bicarbonate | Sigma | Cat#S6297 |
| D-(+)-Glucose | Sigma | Cat#G7528 |
| Potassium chloride | Sigma | Cat#P3911 |
| Sodium pyrophosphate decahydrate | Sigma | Cat#221368 |
| Magnesium sulfate | Sigma | Cat#M7506 |
| Calcium chloride dehydrate | Sigma | Cat#C3881 |
| Magnesium chloride hexahydrate | Sigma | Cat#M0250 |
| D(+)-Sucrose | PanReacAppliChem | Cat#131621.0914 |
| Na-pyruvate | Sigma | Cat#P2256 |
| L-Ascorbic acid (sodium salt) | Sigma | Cat#A4034 |
| NBQX disodium salt | Tocris | Cat#1044 |
| D-AP-5 | Tocris | Cat#0106 |
| Picrotoxin | Abcam | Cat#ab120315 |
| Tetrodotoxin citrate | Abcam | Cat#ab120055 |
| CsMeSO₄ | Sigma | Cat# |
| Tetraethylammonium chloride (TTX) | Sigma | Cat#T2265 |
| QX-314 Chloride salt | Alomone Lab | Cat#Q150 |
| HEPES | Sigma | Cat#H4034 |

| Reagent or resource | Source | Identifier |
|---|---|---|
| Adenosine 5'-triphosphate magnesium salt | Sigma | Cat#A9187 |
| Guanosine 5'-triphosphate sodium salt hydrate | Sigma | Cat#G8877 |
| EGTA | Sigma | Cat#3889 |
| Potassium D-gluconate | Sigma | Cat#G4500 |
| Phosphocreatine disodium salt hydrate | Sigma | Cat#P7936 |
| Cesium chloride | Sigma | Cat#289329 |
| *Software and algorithms* | | |
| Ethovision XT 13 | Noldus | https://www.noldus.com/ |
| MATLAB 2018b | Mathwoks | https://www.mathworks.com/ |
| GraphPadPrism 7.0 | GraphPad | https://www.graphpad.com/ |
| MClust 4.0 | David Redish | http://redishlab.neuroscience.umn.edu/MClust/MClust.html |
| Clampex 10.7 | Molecular devices | N/A |
| Clampfit 10.7 | Molecular devices | N/A |
| pCLAMP 10 | Molecular devices | N/A |
| *Other* | | |
| Datal lynx SX | Neuralynx | https://neuralynx.com/ |
| Noldus USB-IO Box | Noldus | https://neuralynx.com/ |
| Dual Fiber-optic Cannulas | DFC_200/245-0.53_2.0_DF_0.7_FLT | http://doriclenses.com/ |
| BrightLine® YFP-A-Basic-000 filter | Semrock | Cat#YFP-A-Basic-000 |
| Fluorescence light source | Olympus | Cat#U-LH100HG |
| 473 nm laser | Crystal Laser | http://www.crystalaser.com/new/bluelaser.html |
| Multiclamp 700B | Molecular devices | N/A |
| Digidata 1550 | Molecular devices | N/A |
| Image-J | FIJI | https://fiji.sc/ |
| AAV2/9-S5E2-EGFP-IRES-Connexin-36-HA | | |
| AAV2/9-hSyn1-Flex-connexin-36-shRNA1 | | |
| AAV2/9-hSyn1-Flex-scramble | | |
| AAV9-EF1-mCherry-U6-LoxP-Stop-GFP-LoxP-connexin-36-shRNA2 | Vector Biolab | shAAV-260146 |
| AAV9-EF1-mCherry-U6-LoxP-Stop-GFP-LoxP-scramble | Vector Biolab | |
| AAV9-S5E2-Cre | | |

**Animals.** To generate Pv-Cre; WT/*Shank2−/−* mice, *Shank2+/−* mice lacking exons 6 + 7[15] were crossed with Pv-Cre mice (#8069; Jackson Laboratory)[79]. Heterozygous Pv-Cre male 30 mice over the age of 12 weeks were used for all the experiments except for the patch experiment shown in Fig. 3h–k, where we used female mice, partly based on the previous report that male and female Shank2-KO mice show comparable levels of behavioral abnormalities[15]. Experimental

procedures were approved by the Committee on Animal Research at Korea Advanced Institute of Science and Technology (KAIST; approval number KA2016-30). Mice were housed and bred at the mouse facility of KAIST and maintained according to the Animal Research Requirements of KAIST. All animals were fed ad libitum and housed under 12-h light/dark cycle (light phase from 1 am to 1 pm) at 21 °C and under 50–60% humidity. All behavioral experiments were performed during dark-cycle periods.

**Linear chamber test.** The linear chamber test, modified for single-unit measurements from the original three-chamber social interaction test[80], was performed as described previously[48]. Each round of the linear chamber test consisted of 5 sessions (without inter-session delays), including a 5-min habituation session on the platform, 10-min empty–empty, social–object, and object–social interaction sessions, and the last 5-min session on the platform. In the first S–O session, social and object targets were randomly placed on the left and right sides of the apparatus. A novel object and a novel 129 male mouse were used in every new round of the experiments. After each round of the experiments was followed by at least 3 days of isolation. The linear chamber test with an optogenetic stimulation experiment was performed with the inter-round interval of a week to minimize the impacts of blue-light optogenetic stimulation.

**Single-unit recording and analysis.** Single-unit recordings in the mPFC of behaving WT and Shank2−/− male mice engaged in social interactions in the linear chamber test were performed as described previously[48]. Microdrives (modified Flex drive from Open Ephys; a kind gift from Dr. Min Whan Jung) and EIB-36 (Neuralynx, USA) carrying eight tetrodes were implanted in the mPFC of the right and left hemispheres (4 tetrodes on each hemisphere; 1.7 mm anterior and 0.3 mm lateral from bregma and 1.8–2.5 mm ventral from brain surface) under ketamine and xylazine anesthesia. After allowing mice 1 wk to recover from surgery, tetrodes were gradually lowered, and unit signals were recorded from the prelimbic region of the mPFC. Unit signals were amplified 10,000x, filtered between 600 and 6000 Hz, digitized at 32 kHz, and stored in a personal computer using a Cheetah data-acquisition system (Neuralynx, USA). Tetrodes were lowered by 62.5 μm after each daily experiment so that different units could be recorded across successive experiments. The anatomical location of each recorded unit was determined based on the location of electrolytic lesion marks and histological examination of 40-μm coronal brain sections under a multi-slide scanner (Axio scan, Carl Zeiss) and the advancement history of the corresponding tetrode. Behavior and neuronal firing were analyzed as described previously[48].

Valid trials were defined as those with a time interval >2 s from the previous interaction and where the least duration of interaction was >1 s from start to end of nose-poke time. Valid trials were analyzed by detecting nose point from the recorded video file by EthoVision 13.1, and were confirmed by manual corrections for incorrect detection. The mean-firing rates of valid trials for empty, social, and object target interactions were calculated. The total number of neurons recorded and valid trials were as follows: n = 447 neurons from 5 mice [WT] and 472 neurons from 5 mice [KO]; after filtering the neurons with firing rates >0.5 Hz and with activities in all six sessions (E–E, S–O, and O–S), the numbers of neurons were reduced to 295 and 409 neurons, respectively. Numbers of nose-poke trials were as follows: 1042 [WT-E], 916 [WT-S], 851 [WT-O], 1852 [KO-E], 1291 [KO-S], and 1382 [KO-O]).

Target-specific neuronal responses to main factors (target [empty, object and social] and sideness [right and left]) were defined based on two-way ANOVAs followed by Tukey's multiple comparison test. Social-specific neuronal responses were defined as those that showed significant ($p < 0.05$) discriminative discharges for both empty–social and object–social comparisons. The information number for a single neuron was defined as the number of target pairs (empty–social, object–social, empty–object, and left–right) that were significant ($p < 0.05$).

During the rest and each interaction condition (rest, empty, subject, and social), the inter-spike-intervals (ISIs) of putative excitatory and inhibitory neurons were measured. The inter-spike-interval of <200 ms was considered for the spike pattern analysis. In each neuron, a group of spikes with ISI <12 ms was defined as a "high-frequency burst firing" pattern, and the number of spikes in it was counted to estimate the proportion of burst firings in the entire firing activity during each interaction session. The tests were performed using threshold values from 3 to 30 ms for defining the condition of burst firing. For the comparison of the proportion of burst firing, spike timing was randomly shuffled in each interaction, and the proportion of burst firing was calculated from the shuffled spike timing, to compensate for the different number of spikes (firing rate) in each group. The discrimination index d′ was defined by $d' = (\mu1 - \mu2)/\sqrt{1/2\sigma1^2 + \sigma2^2)}$, where $\mu$ and $\sigma$ represent mean and standard deviation of neuronal firing, respectively[52].

**Single-unit recording with optogenetic stimulation.** Mice older than 12 wk were used for single-unit recordings combined with optogenetic stimulation. For each round of social interaction in the experiment, a novel inanimate object and a novel social target (age-matched, male 129/Sv mouse) were used. Optic fiber insertion and microdrive/EIB implantation were performed as described above, with the distance between optic fiber tips and tetrode tips maintained at <500 μm. AAV-DIO-ChR2-EYFP or AAV-DIO-EYFP virus (0.5 μL) was bilaterally injected into

the mPFC followed 1 week later by implantation of a chronic drive with an optic fiber. Tetrodes were lowered by 62.5 μm after each round of experimentation with fewer than ten repetitions. Movement restriction imposed by the weight of the Microdrive, optic fiber, and tether was prevented by attaching a balloon filled with He gas to the tether by a string. Throughout the experiment, mice were stimulated with 1.5-mW blue laser light at 10 Hz for 5 ms. Experiments were repeated every 7 days to minimize the impacts of optogenetic excitation on mouse brain function and behavior. Optogenetically tagged Pv neurons were defined by those that show statistically stronger responses during the first 20 ms relative to the last 20 ms during the 100 ms inter-light-pulse intervals using paired t-test. In vivo experiments testing for the effects of ChETA and SSFO on firing rates of pyramidal neurons were performed using freely moving mice in homo cages with randomized light-on and light-off stimulations. The total number of neurons recorded and valid trials were as follows: n = 316 neurons from 7 mice [WT-EYFP], 220, 8 [WT-ChR2], 232, 4 [KO-EYFP], and 531, 9 [KO-ChR2]; after filtering the neurons with firing rates >0.5 Hz and with activities in all six sessions (E–E, S–O, and O–S), the numbers of neurons were reduced to 194, 72, 163, and 364 neurons, respectively; numbers of nose-poke trials = 390 [WT-EYFP-E], 339 [WT-EYFP-S], 261 [WT-EYFP-O], 271 [WT-ChR2-E], 131 [WT-ChR2-S], 166 [WT-ChR2-O], 957 [KO-EYFP-E], 629 [KO-EYFP-S], 631 [KO-EYFP-O], 920 [KO-ChR2-E], 518 [KO-ChR2-S], and 493 [KO-ChR2-O]).

**Slice electrophysiology.** Mice were anesthetized with isoflurane and decapitated. After brain dissection, coronal slices containing the medial prefrontal cortex (300 μm) were cut in ice-cold oxygenated sucrose-based cutting solution containing (in mM): 75 sucrose, 76 NaCl, 2.5 KCl, 25 NaHCO₃, 25 glucose, 1.25 NaH₂PO₄, 7 MgSO₄, 0.5 CaCl₂ with pH 7.3, equilibrated with 95% O₂ and 5% CO₂, using a vibratome (7000smz-2, Campden Instruments), and then recovered in the same solution for 30 min at 33–34 °C. Slices were then transferred to an incubation chamber filled with oxygenated artificial cerebral spinal fluid (ACSF) containing (in mM): 124 NaCl, 2.5 KCl, 1.3 MgCl₂, 2.5 CaCl₂, 1.0 NaH₂PO₄, 26.2 NaHCO₃, 20 glucose with pH 7.4 at room temperature and slices were kept in <7 h before recordings.

For patch-clamp recordings, slices were transferred to a recording chamber perfused with oxygenated ACSF at 30–32 °C controlled by a peristaltic pump. Neurons in the prelimbic region (layer 2/3) of the mPFC were visualized using ×80 and ×20 objectives (Olympus) simultaneously placed on the stage of upright microscope (BX61W1, Olympus) equipped with infrared differential interference contrast optics in combination with a digital camera (AquaCAM Pro/S3) and also equipped with EMCCD camera for fluorescence imaging (iXon3, ANDOR). Patch microelectrodes were pulled from borosilicate glass (OD 1.5 mm, ID 1.10 mm, WPI) on a Flaming-Brown micropipette puller model P-1000 (Sutter Instruments). Patch microelectrodes had a resistance of 4–8 MΩ. Signals were recorded using a patch-clamp amplifier (Multiclamp 700B, Molecular Devices) and digitized with Digidata 1550 A (Molecular Devices) using Clampex software (Molecular Devices). Signals were amplified, filtered at 2 kHz, and sampled at 10 kHz. Pyramidal neurons were identified by large apical dendrites, and Pv neurons were identified by EYFP fluorescence expression.

In current-clamp recordings, membrane potential was held at −70 mV with intracellular solution (in mM): 135 K-gluconate, 7 NaCl, 10 HEPES, 0.5 EGTA, 2 Mg-ATP, 0.3 Na₂-GTP, 10 Na-phosphocreatine with pH 7.3 and 295 mOsm. Current-clamp experiments were recorded 5 min after obtaining whole-cell configuration. For intrinsic excitability recording, Action potentials (APs) were generated by injecting 500 ms current steps from 0 to 500 pA increasing by 50 pA. For PN output spiking in response to stimulation of all synaptic inputs in extracellular matrix, repetitive electrical stimulation pulses are delivered at 20 Hz frequency. For mefloquine hydrochloride (Tocris) treatments, mefloquine at 25 μM concentration was bath applied at least 20 min to block gap junctional activities during slice recording. For D-cycloserine (Tocris) treatments, D-cycloserine at 20 μM concentration was bath applied at least 10 min to activate NMDA receptor activities during slice recording. During mefloquine or D-cycloserine treatments, all recordings performed in the presence of optogenetic stimulation of Pv neurons (directly stimulated and neighboring) at 10 Hz (473 nm, 5 ms, 12 epochs) under pre-drug (vehicle) and post-drug (mefloquine or D-cycloserine) conditions to obtain paired data.

In voltage-clamp recordings, membrane potential was kept at −70 mV for spontaneous AMPAR-mediated EPSCs (sEPSC) with the following intracellular solution (in mM): 135 CsMS, 10 CsCl, 10 HEPES, 0.2 EGTA, 4 Mg-ATP, 0.4 Na₂-GTP with pH 7.3 and 295 mOsm. Spontaneous GABA$_A$R mediated IPSCs (sIPSC) were recorded at −60 mV membrane holding potential with high chloride intracellular solution (in mM): 150 CsCl, 2 MgCl₂6H₂O, 0.1 CaCl₂2H₂O 10 HEPES, 1 EGTA, 2 Na-ATP, 0.4 Na-GTP. Neurons were voltage-clamped at each holding potential with liquid-junction potential correction of an estimated 10.5 mV. During sEPSC recording, to block inhibitory synaptic responses, 20 μM SR95531 (GABAzine, Tocris) was bath applied but for sIPSC recording, 10 μM NBQX and 50 μM AP-5 (Tocris) were bath applied to block excitatory synaptic responses.

To measure the NMDA to AMPA ratio, holding potential was alternated between −70 mV and +60 mV to isolate AMPAR-mediated EPSCs and NMDAR-mediated EPSCs, respectively. EPSCs were evoked by stimulating the region around

layer 2/3 pyramidal neurons in the prelimbic region of the mPFC with a bipolar electrode at 0.2 Hz. Recordings were performed in ACSF containing 20 μM SR95531 (GABAzine, Tocris) and with the following intracellular solution (in mM): 130 CsMs, 10 TEA-Cl, 10 HEPES, 10 EGTA, 4 Mg-ATP, 0.3 Na2-GTP, 0.5 QX-314 with pH 7.3 and 295 mOsm. For the calculation of the NMDA to AMPA ratio, the mean value of NMDAR EPSCs measured at 40 ms after the onset of stimulation was divided by the mean peak amplitude of AMPAR EPSCs.

For slice electrophysiology with optogenetic stimulation, ChR2-EGFP was selectively expressed in Pv neurons by infecting *Pv-Cre;Shank2fl/fl* mice with AAV-DIO-ChR2-EGFP for ~10 days followed by acute preparation of coronal slices containing the prelimbic region of the mPFC. Pv neurons for optogenetic stimulations were identified by EYFP fluorescence signaling. Blue-light (470 nm) LED illuminations were given by 5-ms pulses at 10 Hz (12 epochs) or 2-ms pulses at 40 Hz (48 epochs) or 80 Hz (48 epochs) with the light intensity of 1–2 mW using a Digital Mirror Device based pattern illuminator (a custom-built Mightex Polygon 400, Mightex Systems) to activate single Pv neurons (neuron cell body, soma only). To make recordings from electrically coupled neighboring Pv neurons, we first selected a Pv neuron in the center field of view that has several other (2–4) surrounding Pv neurons and then established a whole-cell configuration. A neighboring Pv neuron to simulate was next selected based on the following criteria: (1) under the IR-DIC (infrared differential interference contrast) view, the cell body membrane of the Pv neuron should be clearly visible under a moderate contrast condition; (2) under epifluorescence illumination, a large multipolar cell body should be visualized with positive ChR2 expression (should not be saturated); (3) a candidate Pv neuron in the depth of ~300 μm in slices should not overlap with other Pv neurons across slice depth to minimize exciting two or more Pv neurons simultaneously; (4) a candidate Pv neuron should be located deeper than ~20–30 μm from the surface of the slice to ensure the intact morphology of the neuron. Once a pair of Pv neurons is determined, the neighboring Pv neuron was then stimulated with blue light illuminating only the soma (not the surrounding dendrites; 1–2 mW per Pv soma), which was combined with current-clamp recordings of the whole-cell-patched Pv neuron. Only one pair of Pv neurons (stimulated and recorded) was used for each slice experiment. For recordings in single Pv neurons that are directly stimulated, the light was illuminated onto a Pv neuron (1–2 mW per Pv soma) in a pre-established whole-cell configuration. To determine the effects of D-cycloserine and mefloquine on burst firing in neighboring Pv neurons, D-cycloserine (20 μM) and mefloquine (25 μM) were bath applied in brain slices at least for 10 min and 15 min, respectively. Neurons with holding-current changes exceeding 500 pA, or series resistance (Rs) changes >15%, during recordings were discarded from the analysis. Recordings were made in a sequential manner, linking pre and post-drug-treatment sessions with the central drug treatment session. To determine if synaptic transmissions contribute to burst generation, synaptic blockers were bath applied during slice recordings: GABAzine (SR95531; 10 μM) for GABA_A receptor, APV (50 μM) for NMDA receptors, and NBQX (10 μM) for AMPA receptors.

**Electrophysiology analysis.** Current-clamp recordings were analyzed with a custom-built LabView program. To evaluate intrinsic excitability, 500-ms-long depolarizing currents were injected from −200 to 500 pA with increments of 50 pA, and the mean-firing rate was calculated based on the number of evoked action potentials (APs) in response to a depolarizing current injection. The input resistance ($R_{in}$) was determined by measuring the difference between the baseline and the steady-state (post-sag) $V_m$ deflection generated from a −100 pA square current. The membrane capacitance ($C_m$) was calculated by dividing the time constant ($t$) from the exponential curve fitting. For the analysis of single APs, brief depolarizing currents with a duration of 50 ms were injected in neurons with increments of 10 pA until the first AP fires. The rheobase current was defined the minimum current magnitude required to generate an AP. The voltage threshold of AP was defined by measuring the membrane potential at which its 1st derivatives exceeded 5 mV/ms. The differences between the AP threshold and the positive, and the negative peak of the trace were defined as the AP amplitude and the AHP amplitude, respectively. To analyze firings evoked by optogenetic stimulation of Pv neurons, every epoch (12 epochs in 10-Hz and 48 epochs in 40- or 80-Hz stimulation) was individually evaluated and then averaged in a single Pv neuron. Time window in each epoch was calculated as 100, 25, and 12.5 ms for 10, 40, or 80 Hz optogenetic stimulation, respectively. All firings over −20 mV were counted as a threshold. Burst firing was characterized as a series of more than two consecutive spikes with <12 ms inter-spike interval (ISI). The burst ratio (%) was defined as the total number of bursting epochs over the total epochs in a single Pv neuron. The burst duration was defined by the time interval between the first and last burst spikes. For voltage-clamp recordings, sIPSCs were analyzed using Mini Analysis (Synaptosoft) and a custom-made miniature event analysis program (https://github.com/parkgilbong/Minhee_Analysis_Pack). sIPSC events were detected as those with >15 pA amplitude with smoothing parameters of traces (polynominal order for 3 and side points of 35) during 5-min recording in a single neuron. All events starting from the first event up to maximal 100 events (few neurons exhibit less 100 events in sIPSC recording with high KCl internal solution) were averaged to determine amplitude and frequency in a time sequence.

**Leaky integrate-and-fire neuron model.** To perform the simulations of the spiking patterns of Pv neurons, a network of leaky integrate-and-fire (LIF) model neurons is implemented[81]. The network consists of 100 LIF inhibitory neurons that are randomly connected to each other with inhibitory synapses and bidirectional electrical couplings. The probability of synaptic connections between two inhibitory neurons was fixed as 50%. The number of electrical connections per neuron is randomly selected from a Gaussian distribution with the mean $M$ and SD $M/3$, where $M$ varied from 1 to 25.

The subthreshold membrane voltage is calculated from the differential equation:

$$\tau \frac{dV}{dt} = g_L(V_{rest} - V(t) + I_{inh} + I_{gap} + I_{ext}(t) + I_{noise})$$

where $V$ is the membrane voltage, $\tau$ is the decay time constant, $g_L$ is the leak conductance, $V_{rest}$ is the resting potential, and $I$ is the current. $I_{inh}$ is the inhibitory synaptic current, $I_{gap}$ is the gap junction current, $I_{ext}(t)$ is externally given current and $I_{noise}$ is the noise current sampled from a Gaussian with a zero mean and SD σ.

The inhibitory synaptic current, $I_{inh}$ is given by

$$I_{inh}(t) = g_{inh}(t)(V(t) - V_{inh})$$

where $g_{inh}(t)$ represents an "open-and-decay" type synaptic conductance defined as:

$$g_{inh}(t) = g_{max} e^{-\frac{t - t_{spike}}{\tau_{syn}}} \Theta(t - t_{spike})$$

$t_{spike}$ represents a spike timing and $\tau_{syn}$ is a time constant. $\Theta(t)$ is the Heaviside step function.

The gap junction current[82], $I_{gap}$, is given by

$$I_{gap,i}(t) = -\sum_j J_{ij} * F(V_i - V_j)$$

where the connectivity matrix, $J_{ij}$, is defined as

$$J_{ij} = \begin{cases} g_{gap}, & \text{when neurons } i \text{ and } j \text{ are connected} \\ 0, & \text{otehwise} \end{cases},$$

$g_{gap}$ varies from 0 to 0.5, and F(V) is the 10th order Infinite impulse response (IIR) low-pass (<30 Hz) filter.

To mimic laser stimulations to PV neurons in experiments, the external current, $I_{ext}(t)$, is given by

$$I_{ext}(t) = \begin{cases} c, & \text{if mod}(t, 100) = 0 \sim 5\,\text{ms} \\ 0, & \text{otherwise} \end{cases}$$

When the membrane potential exceeds a threshold $V_{th}$, an action potential is generated and the membrane voltage resets to $V_{reset}$ within the refractory period of 1.5 ms.

Two conditions of laser stimulation are simulated: (1) When a single neuron is stimulated: spikes are measured from the stimulated neuron and from neurons connected to the stimulated neuron through gap junction. (2) When all neurons are stimulated: spikes are measured from all neurons. Spike patterns induced by the laser stimulation is categorized into three groups: (1) No fire—when no spike is generated, (2) Tonic—when a single spike is generated by a single laser stimulation, and (3) Burst—when multiple spikes are generated by a single laser stimulation. The probability of each condition is calculated from the activity during 2 s of simulation. The same simulation was also performed with the network consists of 50 LIF inhibitory neurons, to test the effect of different inhibitory neuron density. Custom written MATLAB scripts were used to simulate the model network of neurons with parameters used for simulation as follows: $\tau = 1$ ms, $V_{rest} = -65$ mV, $g_L = 0.4$ mS, $\sigma = 0.2$, $g_{max} = 0.03$ mS, $g_{gap} = 0 \sim 0.5$ mS, $\tau_{syn} = 10$ ms, $c = 0.7$ mA

**Analysis of spike-wave synchrony.** To measure whether the spike firing occurs at a certain phase of the LFP and how optogenetic stimulation changed the phase in WT and *Shank2−/−* mice, the spike timing *tspk* during resting or interaction was isolated. To measure the phase of LFP in each oscillation band, we first filtered the raw LFP signal with 200th-order FIR bandpass filter (Delta: 1–4 Hz, Theta: 5–11 Hz, Beta: 13–30 Hz, Slow Gamma: 30–55 Hz, Fast Gamma: 80–110 Hz). Then, to measure the phase of filtered LFP, we calculate Hilbert transform of the signal X(t),

$$H|X(t)| = x(t) + iy(t)$$

where $X(t)$ is the filtered LFP signal.

Then, the phase at the spike timing *tspk* is calculated as

$$\theta(tspk) = \text{rarctan}(y(tspk)/x(tspk))$$

From the phase distribution of total neuron spike timing when the mouse interacts with empty, object, or social target, the strength of phase locking was estimated from the amplitude of averaged phase vector (unit vector length). The strength of phase locking was calculated using the CircStat function in MATLAB.

**Optogenetic rescue of social interaction**. Mice older than 12 weeks, a developmental stage after full maturation of Pv-positive neurons[83], were anesthetized with ketamine and xylazine and fixed on a stereotaxic apparatus (Kopf). Viruses (0.5 μl of AAV2/9-EF1a-DIO-hChR2(H134R)-EYFP, AAV2/9-EF1a-DIO-SSFO (hChR2-C128S/D156A)-EYFP, AAV2/9-EF1a-DIO-ChETA-EYFP, or AAV2/9-EF1a-DIO-EYFP; UNC vector core) were bilaterally injected (total volume, 1 μl) into the mPFC of both hemispheres (anterior +2.15 mm, lateral ±0.3 mm from the bregma, 1.75 mm from the skull). Then, a metal ferrule with dual optic fibers (0.7-mm gap and 2-mm length; Doric lenses, Canada) was inserted into the injection site with ferrule stabilization using three skull screws; dental cement was then applied to fix the ferrule in place. After 3 weeks for recovery and virus expression, mice were cohoused with their siblings. After behavioral experiments, brains were harvested and fixed in 4% paraformaldehyde. Brain slices prepared using a vibratome (VT1000, Leica) were assessed by post hoc histological evaluation using a confocal microscope (LSM780, Carl Zeiss). Only the behavioral results from mice with proper virus expression and optic fiber targeting were included in the analysis.

**Three-chamber test**. The three-chamber test was performed as described previously[73], except for using a three-chamber apparatus with a large portion of the wall above the doors were removed. The three-chamber test consisted of three 10-min sessions. In the first session, mice were allowed to explore the whole apparatus for habituation. In the second session, mice were allowed to explore an inanimate object or stranger mouse trapped in a small cage inside chambers. In the third session, the object was replaced with a new stranger. A total of three rounds of three-chamber tests were performed for each mouse. The first test performed in the absence of optogenetic light stimulation was to confirm that virus injection and optic fiber implantation do not affect social preference. The second test was performed in the presence of light stimulation. Blue-light stimulation (473 nm, 10 Hz, 5 ms, 1.5 mW for both AAV-EF1a-DIO-ChR2 and AAV-DIO-EYFP groups; 40 Hz, 2 ms with 6.5 mW for AAV-EF1a-DIO-ChETA-EYFP; 2 s 8 mW continuous at the beginning of each session for AAV-DIO-SSFO) was given during the whole three sessions. The third test was performed to test the possibility of any long-lasting effects of light stimulation in the second test. Mice were isolated for 4 days before and between the first three-chamber test (s). The 129/SV strain was as a stranger mouse.

**Open-field test**. Mice were allowed to explore an empty square box (40 × 40 × 40 cm) for 25 min, with alternative 5-min light-off and light-on sessions. In addition to the distance moved, the time spent in the center region of the open-field arena (20 × 20 cm) was measured to determine anxiety-like behaviors. Mouse movements were analyzed using EthoVision XT 10 (Noldus).

**Immunohistochemistry**. Male mice were transcardially perfused with a heparin solution and 4% paraformaldehyde. After the overnight post-fixation, brains were slowly and carefully sectioned in the thickness of 50 μm with a vibratome (VT1000, Leica). Pv (MAB1572, Millipore) and Som (T-4547, Peninsula lab) antibodies were used to mark each interneuron sub-cell types. Stained slices were imaged using a confocal microscope (LSM780, Carl Zeiss). The images were separated by each color channel and converted to 8-bit images, followed by the identification of the brain regions using the Allen Brain Atlas. Image-J program was used to measure and calculate the total number of cell bodies in the anterior cingular (Cg1), pre-limbic (Prl), and infralimbic (IL) areas. Signal thresholding was performed using the same parameters for all images. Cell numbers were calculated using automated particle analysis in Image-J software.

**Mefloquine infusion and connexin-36 knockdown**. For direct infusion of mefloquine infusion into the mPFC of WT and Shank2-KO mice, a canulae (Plastics Ones) was inserted bilaterally into the mPFC region of the mouse brain (ML ±0.3 mm, AP + 1.9 mm, depth 1.5 mm) on the first day and stabilization of 7 days. On the day of experiment, mice received mefloquine or vehicle in the mPFC through the inserted cannula (0.5 μl for each hemisphere, 25 μM concentration), followed the three-chamber social-interaction test after 50-min rest. For connexin-36 knockdown experiments, two independent connexin-36 knockdown viruses (shRNA1 [homemade] and shRNA2 [Vector Biolab]) were used. For shRNA1 virus, shRNA constructs (targeting the nt 503–521 of sequence of the NM_010290.2 sequence, respectively) were subcloned into AAV-hSyn1-Flex-shRNA. AAVs were injected into the mPFC of WT mice (ML ± 0.3 mm, AP + 1.9 mm, depth 1.5 mm) together with AAV-S5E2-Cre for shRNA1 and shRNA2, followed by 2-week rest for shRNA expression and behavioral tests. AAV-hSyn1-Flex-connexin-36-shRNA1 was validated using HEK293T cells expressing a mouse connexin-36-HA construct (Origene). pAAV-S5E2-EGFP-IRES-Connexin-36-HA and EGFP-IRES-Connexin-36-HA were synthesized by overlapping PCR and subcloned into pAAV-S5E2-GFP-fGFP vector (Addgene #135631) by replacing the GFP-fGFP using NheI and EcoRV restriction enzymes. AAV9-EF1-mCherry-U6-LoxP-Stop-GFP-LoxP-connexin-36-shRNA2 was described to induce ~80% knockdown of connexin-36 in cultured cells (Vector Biobabs; shAAV-260146).

**Reporting summary**. Further information on research design is available in the Nature Research Reporting Summary linked to this article.

## Data availability
All data supporting the findings of this study are provided within the paper and its Supplementary Information. All additional information will be made available upon reasonable request to the authors. Source data are provided with this paper.

## Code availability
The code for the custom-made miniature event analysis program is available at https://github.com/parkgilbong/Minhee_Analysis_Pack.

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

## Acknowledgements

Drs. Dohoung Kim and Jeoung-Wook Ghim kindly provided customized Microdrive and EIB-32. This work was supported by the Basic Science Research Program through the National Research Foundation of Korea (NRF) funded by the Ministry of Science and ICT (NRF-2019R1A2C4069863 to S.P.; 2018R1A5A2025964 to S.J.K.), NRF-2019R1A2C1084812 (to E.L.), IBS-R001-D2 (to J.M.P), IBS-R002-D2 (to M.W.J.), and IBS-R002-D1 (to E.K.).

## Author contributions

E.L. and E.K. planned experiments; E.L., I.R., W.K., I.J.C., and H.S. performed in vivo recording experiments; E.L., W.C., I.R., W.K., H.S., and I.J.C. analyzed single-unit data; W.C. and S.P. designed computational model simulations; W.C. performed model

simulations; E.L., S.L., T.L., Jisoo K., Y.W.N., and S.Y.L. performed optogenetic rescue experiments and local infusion-related experiments; C.C., Jihye K., S.H., R.K., T.Y., Y.E.Y., H.S., and J.J.S. performed electrophysiological experiments; S.L. performed immunohistochemistry experiments; E.L., K.D., S.J.K., J.M.P., M.W.J., S.B.P., and E.K. wrote the manuscript.

## Competing interests

The authors declare no competing interests.
