## [Peer Review File · Nature Communications]

Reviewers' Comments:

Reviewer #1:

Remarks to the Author:

The manuscript by Lee et al. titled "Excitatory synapses and gap junctions cooperate to improve Pv neuronal burst firing and cortical social cognition in Shank2-mutant mice" presents in vivo and in vitro optogenetic and electrophysiological recording studies in the context of social interaction. The conclusion they tried to draw regarding to Shank2 mutant mice, loss of NMDAR function in PV neurons and gap junction coupling and their effects on reduced social interaction are difficult to follow. The authors over-interpreted their data and there are several logical gaps. Additional experimental approaches are required to obtain a conclusive interpretation.

Prior work has implicated a negative effect of NMDAR on gap junction coupling (Nature neuroscience 8, 1720-1726, 2005). In contrast, the authors propose here that NMDAR promote gap junction coupling. The Shank2 mice (global KO and PV-specific KO) seem to respond to NMDAR differently from WT in terms of gap junction coupling, but this needs further experiments to clarify the difference.

1) Figures 1-2 are in vivo recording experiments carried out in the global Shank2 KO mice, followed by in vitro experiments in Fig. 3f- onward carried out in PV-Cre/Shank2-flox mice to dissect PV neuron properties. However, the authors have not yet demonstrated that PV-Shank2 mice display the same social interaction deficits as in global knockout. It would be important show that the deficits presented in Figure 1c for the global Shank2 KO are reproduced in PV-Cre/Shank2-flox mice (Figure 7h for PV-KO should be compared side-by-side to WT currently in Supplemental Figure 12c and two-way ANOVA statistical analysis carried out between genotypes testing for possible interactions with treatment, Chr2 or eYFP). Also, in vivo recordings from the Shank2 global KO shown in Figure 1b should be repeated in PV-Cre/Shank2-flox mice. Another very important issue is whether PV-Cre/Shank2-flox mice have any olfactory deficit (which itself would strongly affect social interaction).

In supplemental Figure 12c, the proportion of neurons (%) for target column for Chr2 does not add up for the side, S, O and E combined.

2) From Fig 1, the baseline activity in KO before nose-poke is much higher than WT. Does this suggest reduced evoked PV-mediated Gabaergic output onto PN (consistent with Fig. 3 f, g, h) but not (Fig. 3i, j)?

3) The authors say there is a PV neuron deficit, but one could interpret their data to mean that there is a PV neuron hyper-activity: overlaying figures 3g and 3h show that wt is always above Shank2 KO, so at baseline Shank2 KO look like they have partially reduced firing rate of pyramidal neurons at different "simulated" EPSC. This suggests increased PV neuron function at baseline.

On the other hand, Figure 3i and 3j suggest that PV neuron function is impaired in the KO compared to WT, since the spike probability of pyramidal neurons (PN) is not reduced by photoactivating PV neurons. The authors stated that there is a trend toward lower numbers of PV neurons in Shank2 global KO in Sup Fig 6, but there is no comparison of PV neurons number in PV-Cre/Shank2-flox mice.

Together, the results in Figure 3 g, h and I, J appear contradictory.

4) It is ambiguous what the authors mean by "simulated" EPSC in Fig. 3g and 3h. The authors need to explain it in the manuscript or online methods. Simulate means to imitate, while stimulate is to evoke. On page 11, line 264, "simulated" should be "stimulated".

5) Figure 4 is straightforward and supports the notion of frequency-dependent coupling in the PV-Shank2 KO.

6) Figure 5 shows that the gap junction blocker mefloquine has a more profound effect in PV

Shank2 KO, eliminating burst firing. However, mefloquine had no effect on burst firing in the WT mouse, suggesting that burst firing is not an index of PV neuron coupling in WT mice. How do the authors explain this difference?

7) In Figure 6C, D-cycloserine is said to stimulate NMDAR-dependent gap junction coupling in PV-Shank2 KO associated with increased burst activity in neighboring PV neurons. How do the authors explain why DCS increases gap junction coupling in KO but in WT it has the opposite effect and reduces coupling? It is essential that the authors combine DCS with mefloquine (Cx36 blocker) and demonstrate that mefloquine can block the effect of DCS in PV-Shank2 KO mice on burst activity.

The observation of NMDAR activity increasing coupling of gap junctions goes against prior literature showing NMDAR REDUCE gap junction coupling (ref 63). How do the authors reconcile this difference?

The author should also test the effect of blocking NMDAR on gap junction coupling in WT and in Shank2 mutant mice.

8) The global KO of Shank 2 shows no difference in Cx36 expression with age-matched WT mice, the authors should also compare Cx36 expression in the PFC of PV-Shank2 KO mouse?

9) Previously, this group published in Nature in 2012 that the global Shank2 KO mice display ASD-like social deficits that could be rescued by a positive allosteric modulator (PAM) of mGluR5, known to facilitate NMDAR signaling. Now, the authors also used PV-Cre mice to ablate a floxed allele of Shank2 specifically in parvalbumin neurons. These mice are also reported to display ASD-like social deficits. The authors should repeat the mGluR5-PAM strategy used in their prior paper to test whether this also works in the PV-Cre/Shank2flox mice. Barnes et al. (Mol Psychiatry 20, 1161-1172, 2015) showed mGluR5 receptors are present on PV neurons and functionally important. Since there are NMDAR on PV neurons, their PAM strategy should also work in the PV-Cre/Shank2flox mice.

10) In their Introduction, page 4 lines 82-84, the authors state: "Whether Pv neurons display burst firing and how Pv neuronal bursts are generated and regulated are not know". However, this is technically incorrect. A prior study by Natlow et al. (Neuron, Vol. 38, 805-817, June 5, 2003) that these authors should cite found that Pv neurons in layer 2/3 of the frontal and somatosensory cortex are multipolar bursting cells. Importantly, this paper showed that gap junction coupling between PV neurons could be increased by carbachol (a muscarinic agonist). This suggests that the NMDAR-dependent uncoupling of PV neurons is not developmentally irreversible but can be modulated in adult brain by muscarinic receptor signaling. Since carbachol increases gap junction coupling, would treating WT mice with carbachol produce similar social interaction deficits in WT mice as in Shank2 KO mice?

11) If increased gap junction coupling between PV neurons in PV-Shank2 KO mice contributes to impaired social interaction, how can optogenetic activation of PV neurons rescue social interaction deficits in Figure 7? Won't Optogenetic activation of PV neurons in mutant mice further amplified the burst firing of PV neurons and further impair social interaction?

Additional points:

1) Page 4, line 93, please insert the number of WT and Shank2+/- mice implanted with tetrodes including the coordinate used.

2) Page 4, line 94 or (Fig 1a), what are dissociation intervals by translocation from reset box to exploration apparatus and so on for each social interaction task. There were two repetitive 10-minutes S-O tests in the same paradigm. How are they different? Were data averaged or pooled for these two tests? Was there any adaptive rundown or desensitization for these repeat measurements? Were any other target pairs set for tests, e.g. E-O or S-E? If yes, name them all.

3) page 25, line 584, what does L-R represent? Also please label the pie charts in Fig 1d.

4) Fig 3d, are sEPSC frequency and amplitude different in Pv neurons between WT and Shank2 +/-

as in Fig3a? It is interesting to know since Shank2 $-/-$ mice are implicated with NMDAR hypofunction in Fig6a and b.

5) It seems that the vehicle itself simulated burst and tonic firings, while mefloquine (MQ) suppresses them in WT and KO treated mice (Fig 5b) compared to their non-treated mice (Fig 4b). Two-way ANOVA might be the proper analysis to evaluate burst proportion and firing rates for interaction of gene (WT vs KO) and MQ effects. The same goes for Fig 6c to evaluate the interaction between the gene and the D-cycloserine (DCS) effects by two-way ANOVA.

6) Fig 7h, the conclusion of boosting Pv neuronal burst by 10-Hz light stimulation that improves social representation and behavioral social interaction is relatively weak. The significance for rescue is only shown in 1 of 5 paired bars whereas it was robust for the phenotype in KO (4/5).

Reviewer #2:

Remarks to the Author:

The manuscript "Excitatory synapses and gap junctions cooperate to improve Pv neuronal burst firing and cortical social cognition in Shank2-mutant mice", by Lee et al, proposes an interesting link between firing properties of networked Pv-positive interneurons, which being modulated by NMDA excitatory drive and gap junction connectivity, drive social behavioral dysfunction in a popular model of autism spectrum disorders. The results are novel and reinforce recent evidence on the literature on the effect of interneuron modulation of social behavior. The authors use advanced methodological tools, including genetics manipulation, optogenetics, in vivo neuronal activity recording and whole-cell patch clamping.

The manuscript will be of interest to the broad field of neuroscience as it proposes links on critical aspects of circuit properties to behavioral modulation.

However, there are several issues with data organization, presentation, clarity, as well as a few logical leaps that prevent a strong endorsement of the manuscript in its present form.

Major issues:

A) L103: The authors propose that neurons may show discriminative responses to S, O and E "(Fig. 1b [right]), while others showed no target-specific firing.". However, the indicated graph shows a clear target-specific firing towards the Object (O), with no significant response to E or S. Is this just a poorly chosen example? Also, since both target-specific and broadly-tuned neurons may be found in both genotypes, please show examples of both classes for each genotype to avoid confusion.

B) L104-109 and Figure 1c-d:

I) The authors propose that KO "neurons responded more strongly to the non-social targets". The way the data is presented is hard to interpret since the text, figure and figure legend are inconsistent in their labelling and what is being represented is not clear. For example, Fig 1c mentions, Target, Side, S-spc, O-spc, E-spc... These abbreviations are not mentioned in the text or figure legend.

II) Additionally, the figure legend and text, indicate that the total of target-specific neurons is greater in the KO animals. However, eyeballing the sum of "Side + S-spc + O-spc + E-spc" shows a different number than the percentage indicated under "Target", which is presumably the "total (all combined), targets" as indicated in the Figure legend (L581).

III) Finally, if the authors can clarify the issues above, still, the assertion in L109 that "non-social targets are more strongly represented", should be perhaps interpreted as non-social discriminative neurons being more prevalent in Shank2 $-/-$ mPFC. However, a major caveat is if Shank2 KO mice engage in fewer bouts of social interaction, could it be that there is an undersampling regarding the instances of social nose-pokes in the KO, mice? Can the authors assuage this concern?

IV) It would be useful to indicate the total number of neurons recorded and the number of total trials performed.

V) Figure 1d is also a problem. The point the authors are making is that there is an increase in the total number of targets a single KO neuron responds to. However, the data is very hard to

interpret.

- a. The figure legend mentions 0-4 pairs (i.e., 5 conditions) but then mentions only 4 pairs (S-O; S-E; E-O and L-R).
- b. More problematic is also the reduction of complexity to dyadic pairs. Presumably, since one neuron is recorded from multiple sessions of different S, E, O combinations, a more refined analysis is necessary to extract meaningful information, e.g. network, graph analysis, or correlation matrixes. Some of this analysis is tentatively exposed in Supplementary Figure 2-3, but the cursory description of these data is problematic.

C) Pv vs SOM

- a. A major issue in the author's narrative arrives when confronting Figure 1e-h, Figure 2 and Supplementary Figure 6. Specifically, the authors find differences in burst firing on putative inhibitory neurons (Fig 1), go on to observe abnormal oscillations in Shank2 KO mice mPFC and finally segue into Pv neurons as likely culprits. However, since Pv neurons are only one of the populations of inhibitory neurons in the mPFC this raises significant concerns. Particularly because the authors' data show a very small number of PV neurons <5 cells/mm² (Sup. Figure 6), when compared to SOM neurons (10x more prevalent?!) (see <https://doi.org/10.1002/dneu.20853>). If Sup. Fig 6 is correct, what is the likelihood that the phenotype described in Fig 1g-h is derived from Pv-neuron population?
- b. Can the authors differentiate the inhibitory neurons recorded in vivo segregated as Pv-like or SOM-like neurons? (see <https://doi.org/10.1126/sciadv.aay4073> and <https://doi.org/10.1016/j.cell.2015.11.038>)
- c. Note that the reference indicated in that section (Lim et al. 2018), presumably DOI:10.1016/j.neuron.2018.10.009, is not indicated in the reference list.

D) Regarding the optogenetic stimulation of Pv neurons (Fig 4) the authors find an increase in burst firing when neighboring Pv-Pv linked-neurons are recruited by ChR2 activation. However, a competing hypothesis, is that the rheobase, AP threshold or resting membrane potential of the target cells is already somewhat perturbed due to loss of Shank3. Can the authors provide a table with the membrane properties of the recorded neurons to assuage these concerns?

E) Gap junction and electrical coupling:

- a. L228: The authors indicate that gap junctions may function as low-pass filters to explain the difference between 10 and 40Hz stimulation... However, a common understanding is that this filtering capability is significantly noticeable only when comparing between very high-frequency events (spikes) but perhaps not as much within lower frequency (10-100Hz). Additional experiments should be formed to assess the attenuation in the neighboring neurons using a wider range of frequencies and ChETA or direct current injections.
- b. Another factor pertaining to gap junctions will be a charge diffusion that is more greatly dispersed in more connected cells (in this case the Wt), when compared to less connected cells (Shank2 KO), by virtue of the reduced number of Pv partners in the network of the KOs (Supplementary Figure 6). Can the authors expand their modeling work to estimate the impact of a reduced number of electrically coupled partners or discount the consequences of having a system of similarly expressed gap-junctions (no upregulation) but where the lower partner connectivity putatively leads to greater charge transfer per gap junction.

F) There is a large discrepancy in the firing properties of Wt neurons recorded in Fig 4b (control) and 5d (Vehicle condition) or Fig 6c. The same between Fig 4a and Fig7d (here the KO stimulated has a very different response even in the trace example!). Should these be equivalent preparations in vehicle and control? If yes, what can explain such a significant difference in firing properties (tonic, burst etc), the average firing rate of the neurons, or traces?

G) Finally, a major weakness of the work is that the very interesting assertion that gap junctions are playing a causative role in the bursting activity deficits in Shank2 KO mice, which in turn would lead to social behavioral deficits; falls short in a direct mechanistic test. One of two experiments would be critical to perform:

- a. Further enhance the upregulation of gap junctions (connexin 36?) in a Pv-specific manner to support the authors' theory of a compensatory role of this mechanism, thereby expecting a rescue of the phenotype.
- b. Or, selectively perturb Gap junctions in Pv neurons in wildtype mice to phenocopy social behavioral dysfunction.

Minor points:

Figure 1a –change 2nd S-O to O-S to avoid confusion.

Confirm the statistical analysis is Figure 2c, since a "eyeball analysis" (i.e. close means with SEM overlap) raises some concerns on the comparison between Pre and Post Gamma PSD in the Wt animals showing $p < 0.05$.

Why were only female mice used in the patch experiments? Are the authors confident there are no differences between genders? Can they provide data to this point? This is particularly important in a study that goes from cellular, circuit and behavioral analysis and where a sense of continuum is given.

L576 Figure 1 – Symbol is missing.

Figure 1b Wt graph is repeated in Supplementary figure 1b WT.

Reviewer #3:

Remarks to the Author:

Review of Lee et al Nat Comms 2021:

This manuscript by Lee et al. uses in-vitro and in-vivo electrophysiology together with behavioral testing to investigate the cellular and circuit mechanism of social behavior dysfunction in the Shank2 mouse model of autism. The authors provide compelling evidence for the involvement of both gap junction hyperfunction and NMDA receptor hypofunction in the behavioral phenotypes displayed by Shank2^{-/-} mice. The electrophysiological experiments are well-designed and carefully executed, using a set of paradigms previously developed and validated by the authors. The authors further develop a novel approach for testing the responses of PV neurons to optogenetic stimulation of neighboring PV neurons, showing that differences between the WT and KO mice are prominent in this aspect of cortical physiology. This is an innovative concept which has received little attention in the investigation of the pathophysiological mechanisms of autism. Although the authors do not provide a solid causal link between the gap junction dysfunction and the social behavior phenotypes, this manuscript is exciting for its innovative mechanistic insight, and will be of interest to a broad community of scientists studying the prefrontal cortex, autism spectrum disorders and synaptic physiology.

Major comments:

- Can the differences in burst firing across targets in Fig. 1h (in WT mice) be attributed merely to the differences in firing frequencies between the targets (S vs. R)? Although the average firing rate was not different in the inhibitory neurons (Fig. 1f) it would be good if the authors could show that shuffling the spike times eliminates the burst firing differences.
- Figures 4, 5 and 6 rely on a novel technique, in which the authors stimulate individual PV neurons while recording from the stimulated cell or a neighboring one. This experiment is very interesting, and it allowed the discovery of a novel gap junction phenotype in the Shank2 KO mice. This is a very clever experimental paradigm. However, some technical information is missing in the description of this experiment. For instance, what was the light power density used for these experiments? What was the illuminated area when using the DMD device? Are the authors confident that they are really stimulating only a single PV neuron? A control experiment for this

could be recording in the same configuration with GABA and glutamate synaptic blockers. This will allow quantification of the residual photocurrent triggered directly vs. the gap junction-mediated current. At minimum, the authors should report the distances between the stimulated and recorded neurons and show that these distances are not systematically different between the two genotypes.

- The authors demonstrate that connexin-36 expression is not different between WT and KO mice across a wide range of ages. The fact that PV neurons are slightly less abundant still cannot account for the substantial increase in gap-junction mediated currents in the KO mice. Another potential explanation would be that the arborization of PV neurons in the KO mice is more extensive and this allows for more extensive coupling between cells. Anatomical tracing of dye-filled PV neurons would reveal whether structural differences might underlie the differences in electrophysiology identified by the authors.

- Figure 7: the authors state that PV neuron stimulation "restored social interaction in Shank2^{-/-} mice". However the statistical comparison in Fig. 7 is unclear - is there a significant across-group interaction in the 2-way ANOVA? The indication on the figure only implies a within-group repeated measures effect.

- The authors rightfully claim in their Discussion that their results might indicate an involvement of mPFC gap junction hyperfunction in the behavioral phenotypes observed in Shank2^{-/-} mice. I wonder whether they have tried to directly test this hypothesis by applying mefloquine directly in the mPFC in the KO mice? This might indeed be a direct test of their hypothesis and help consolidate the novel concept introduced in this manuscript, that gap junction hyperfunction is a potential mechanism of autism-related dysfunction and a potential target for future therapies.

Minor comments:

- Line 85 - "cognitions" should be singular.

- Lines 88-91 - sentence with multiple clauses - very difficult to read. Please simplify and avoid multiple clauses throughout the manuscript.

- 595 firing-firing rate?

- Supplementary Fig. 4: instead of "criteria", the term should perhaps be "threshold".

- Line 149 - it seems that theta power did decrease in the mPFC upon nose poke.

- General comment: the authors use the term "nose poke" to describe interaction with the target. This term is traditionally used in operant conditioning to describe a specific action made by the animal. The "nose poke" in that context has a very defined motor pattern and is not similar to the sniffing that mice perform when investigating a conspecific or an object. Perhaps a better term would be "sniffing onset"?

- Line 167: "PV neuronal density tended to decrease strongly in the Shank2^{-/-} mPFC, unlike that of somatostatin-positive neurons" - the word "strongly" seems inappropriate given that none of the comparison reach significance, while somatostatin interneurons are actually significantly reduced in the Cg. This claim should somehow be toned down.

- Figure 4: Do PV neurons in Shank2 KO mice have the same ChR2 photocurrent amplitude as WT PV neurons when stimulated directly? The observed changes might be a simple result of altered translation and membrane trafficking in the KO mice.

- Supplementary information- patch clamp methods: line 121: please check the temperatures stated.

Point-by-point response to reviewers' comments

REVIEWER COMMENTS

Reviewer #1 (Remarks to the Author):

The manuscript by Lee et al. titled "Excitatory synapses and gap junctions cooperate to improve Pv neuronal burst firing and cortical social cognition in Shank2-mutant mice" presents in vivo and in vitro optogenetic and electrophysiological recording studies in the context of social interaction. The conclusion they tried to draw regarding to Shank2 mutant mice, loss of NMDAR function in PV neurons and gap junction coupling and their effects on reduced social interaction are difficult to follow. The authors over-interpreted their data and there are several logical gaps. Additional experimental approaches are required to obtain a conclusive interpretation.

Prior work has implicated a negative effect of NMDAR on gap junction coupling (Nature neuroscience 8, 1720-1726, 2005). In contrast, the authors propose here that NMDAR promote gap junction coupling. The Shank2 mice (global KO and PV-specific KO) seem to respond to NMDAR differently from WT in terms of gap junction coupling, but this needs further experiments to clarify the difference.

→ We sincerely appreciate the thoughtful comments of the reviewer.

1) Figures 1-2 are in vivo recording experiments carried out in the global Shank2 KO mice, followed by in vitro experiments in Fig. 3f- onward carried out in PV-Cre/Shank2-flox mice to dissect PV neuron properties. However, the authors have not yet demonstrated that PV-Shank2 mice display the same social interaction deficits as in global knockout.

→ We did not use *Pv-Cre;Shank2^{f/f}* mice in our study. Rather, we used Pv-Cre mice crossed with Shank2 global knockout mice (*Pv-Cre;Shank2^{-/-}* but not *Pv-Cre;Shank2^{f/f}*) so that the Cre recombinase expressed in Pv neurons can drive the expression of virus-delivered transgenes such as Dio-ChR2-EYFP to visualize or stimulate Pv neurons. Our apologies for the insufficient description in the original manuscript, and we accordingly clarified this in the revised manuscript.

It would be important show that the deficits presented in Figure 1c for the global Shank2 KO are reproduced in PV-Cre/Shank2-flox mice (Figure 7h for PV-KO should be compared side-by-side to WT currently in Supplemental Figure 12c and two-way ANOVA statistical analysis carried out between genotypes testing for possible interactions with treatment, Chr2 or eYFP).

→ Regarding the reproducibility, again, we used *Pv-Cre;Shank2^{-/-}* mice but not *Pv-Cre;Shank2^{f/f}* mice.

Regarding the two-way analysis for the results described in **Fig. 7h** and **Supplementary Fig. 12c**, we now show all the results in the new **Fig. 7h** for better comparisons. However, these neuronal proportions are not raw data and were derived

from a two-way ANOVA-based extraction of target pair-discriminative, or target-specific, neurons using Venn diagram analyses, making it inappropriate to try two-ANOVA analysis. We thus tried Chi-square analysis with Bonferroni correction and multiple comparisons with p value adjustments. We made these aspects clear in the figure legend.

The increased Target- and E-specific neuronal portions in Shank2-KO-EGFP mice (**Fig. 7h**) partly differs from the increased Target-, E-, O- and Side-specific neuronal proportions in naive Shank2-KO mice (**Fig. 1d**), as correctly pointed out by the reviewer. This may be attributable to the different genetic backgrounds of *Shank2*^{-/-} and *Pv-Cre;Shank2*^{-/-} mice (C57BL6/N and C57BL6/N + C57BL6/J, respectively) and AAV-mediated gene delivery in *Pv-Cre;Shank2*^{-/-} mice but not in *Shank2*^{-/-} mice. We clarified it in the main text.

Also, *in vivo* recordings from the Shank2 global KO shown in Figure 1b should be repeated in PV-Cre/Shank2-flox mice.

→ As mentioned above, we used *Pv-Cre;Shank2*^{-/-} mice but not *Pv-Cre;Shank2*^{fl/fl} mice. Again, our apologies for the inconvenience.

Another very important issue is whether PV-Cre/Shank2-flox mice have any olfactory deficit (which itself would strongly affect social interaction).

→ *Shank2*^{-/-} mice have been reported to have normal olfactory function (Won et al., Nature 486:261-265, 2012).

In supplemental Figure 12c, the proportion of neurons (%) for target column for Chr2 does not add up for the side, S, O and E combined.

→ It is because of the way we calculated the neuronal proportions based on the results in the Venn Diagrams, as mentioned above. We clarified this by elaborating the main text, figure legend, and figure panel (**Fig. 1d** showing miniature Venn diagrams).

2) From Fig 1, the baseline activity in KO before nose-poke is much higher than WT. Does this suggest reduced evoked PV-mediated Gabaergic output onto PN (consistent with Fig. 3 f, g, h) but not (Fig. 3i, j)?

→ Our apologies for the confusion. These are merely two examples from hundreds of neurons, and some *Shank2*^{-/-} neurons show increased activity before nose poke while others do not. To minimize the confusion and also in response to the other reviewer's comment, we now show two examples (target-specific and broadly-tuned neurons) for both WT and KO mice (**Fig. 1b**), which is now clarified in the main text.

3) The authors say there is a PV neuron deficit, but one could interpret their data to mean that there is a PV neuron hyper-activity: overlaying figures 3g and 3h show that wt is always above Shank2 KO, so at baseline Shank2 KO look like they have partially reduced firing rate of pyramidal neurons at different "simulated" EPSC. This suggests

increased PV neuron function at baseline.

On the other hand, Figure 3i and 3j suggest that PV neuron function is impaired in the KO compared to WT, since the spike probability of pyramidal neurons (PN) is not reduced by photoactivating PV neurons. The authors stated that there is a trend toward lower numbers of PV neurons in Shank2 global KO in Sup Fig 6, but there is no comparison of PV neurons number in PV-Cre/Shank2-flox mice.

Together, the results in Figure 3 g, h and I, J appear contradictory.

→ As shown below, there is no significant genotype (WT vs. KO) difference in firing rates, suggesting that there is no baseline difference. In line with this, there is no genotype difference in spontaneous inhibitory synaptic input (sIPSC) to pyramidal neurons (**Fig. 3b**). The results shown in **Fig. 3i,j** (now **Fig. 3j,k**) does show a genotype difference in Pv-dependent PN inhibition under light-on (not light-off) conditions, as correctly pointed out by the reviewer. These results collectively suggest that the impaired Pv-dependent PN inhibition occurs under stimulated but not baseline conditions. We clarified this in the text.

4) It is ambiguous what the authors mean by "simulated" EPSC in Fig. 3g and 3h. The authors need to explain it in the manuscript or online methods. Simulate means to imitate, while stimulate is to evoke. On page 11, line 264, "simulated" should be "stimulated".

→ We used this term as it was used in the paper that originally described a similar result (Yizhar et al., Nature 477:171–178, 2011). Briefly, the simulation means simulating the in vivo patterns of Pv stimulation by injecting random-size currents, unlike other stimulation protocols where gradually increasing current sizes are used. We clarified this in the figure legend, and also modified the figure to help understand the random nature of currents injected.

5) Figure 4 is straightforward and supports the notion of frequency-dependent coupling in the PV-Shank2 KO.

→ We appreciate the comment.

6) Figure 5 shows that the gap junction blocker mefloquine has a more profound effect in PV Shank2 KO, eliminating burst firing. However, mefloquine had no effect on burst firing in the WT mouse, suggesting that burst firing is not an index of PV neuron coupling in WT mice. How do the authors explain this difference?

→ To address this interesting comment, we performed additional experiments (Mefloquine treatment in WT mice; the n number was increased from 14 to 29) for more solid conclusions and confirmed that mefloquine has minimal effects on burst firing in neighboring WT Pv neurons (**Fig. 5b**), suggesting that burst firing is unlikely to be a reliable index of gap junctional coupling in WT mice, as correctly pointed out by the reviewer. However, this difference between WT and Shank2-KO mice in mefloquine effects could be attributable to the low baseline gap junctional activity in WT Pv neurons. In support of this possibility, in our new results from the experiment suggested by this reviewer's comment #7 (see below), mefloquine did decrease burst firing in WT Pv neurons when the baseline gap junctional activity and burst firing were elevated by prior D-cycloserine treatment (**Fig. 6d**). We clarified this in the revised main text.

7) In Figure 6C, D-cycloserine is said to stimulate NMDAR-dependent gap junction coupling in PV-Shank2 KO associated with increased burst activity in neighboring PV neurons. How do the authors explain why DCS increases gap junction coupling in KO but in WT it has the opposite effect and reduces coupling? It is essential that the authors combine DCS with mefloquine (Cx36 blocker) and demonstrate that mefloquine can block the effect of DCS in PV-Shank2 KO mice on burst activity.

→ In response, we tested if the DCS-dependent increase in burst firing requires gap junctional activity. We found that gap junctional inhibition by mefloquine (but not vehicle) treatment significantly decreased DCS-dependent burst enhancements in both in WT and KO neighboring Pv neurons (**Fig. 6d**).

Regarding the opposite effect in WT mice, the data in the original **Fig. 6c** actually show that DCS treatment increases burst firing in WT neighboring Pv neurons, similar to the effect of DCS on Shank2-KO Pv neurons, which is consistently observed in the new **Fig. 6c** with larger n numbers. We clarified this in the revised Results.

The observation of NMDAR activity increasing coupling of gap junctions goes against prior literature showing NMDAR REDUCE gap junction coupling (ref 63). How do the authors reconcile this difference?

→ This paper (Arumugam et al., Nat Neurosci 8:1720, 2005) shows a gradual decrease in the expression levels of Connexin-36 in the rat hypothalamus by a chronic MK801-dependent blockade of NMDARs through mechanisms including CREB-dependent modulation of gene expression. We think that our result is an 'acute' effect of NMDAR activation on burst firing, which differs from the 'chronic' decrease in Connexin-36 in the previous paper. The reported chronic effect appears to be a negative feedback to tone

down excessive burst firing induced by NMDAR activity, which increases during development. We commented on these aspects in our Discussion.

The author should also test the effect of blocking NMDAR on gap junction coupling in WT and in Shank2 mutant mice.

→ We appreciate this comment. However, we were discouraged to perform the requested experiment because NMDAR function is already too low in Shank2-KO Pv neurons (**Fig. 6a**) to try an additional inhibition of NMDAR activity. In addition, our new results mentioned above (DCS + mefloquine treatment) demonstrate that NMDARs act synergistically with gap junctions (**Fig. 6d**). However, we did perform a new experiment where all synaptic transmissions (excitatory and inhibitory) were blocked (APV for NMDARs, NBQX for AMPARs, and GABAzine for GABA receptors) in response to the reviewer #3 and found that burst firing is not affected in WT Pv neurons and only moderately decreased in Shank2-KO Pv neurons (**Supplementary Fig. 6b**); the moderate decrease (~2-fold) was in sharp contrast with the mefloquine-dependent decrease (~20-fold) in burst firing in Shank2-KO Pv neurons (**Fig. 5b**). These results suggest that gap junctional activity mainly mediates the enhanced burst firing in Shank2-KO Pv neurons, while NMDARs represent a modulator cooperating with gap junctions.

8) The global KO of Shank 2 shows no difference in Cx36 expression with age-matched WT mice, the authors should also compare Cx36 expression in the PFC of PV-Shank2 KO mouse?

→ As mentioned above, we used *Pv-Cre;Shank2^{-/-}* mice but not *Pv-Cre;Shank2^{fl/fl}* mice. In addition, we have to point out that, while performing additional immunoblot experiments during revision, we found that the connexin-36 protein band shown in **Supplementary Fig. 9** in the original manuscript is not reliable as judged by the immunoblot experiments using connexin-36 knockout brain samples; in vivo endogenous connexin-36 proteins could not be detected in immunoblot analyses despite that we tested multiple antibodies. We thus decided to remove the immunoblot results in the revised manuscript.

9) Previously, this group published in Nature in 2012 that the global Shank2 KO mice display ASD-like social deficits that could be rescued by a positive allosteric modulator (PAM) of mGluR5, known to facilitate NMDAR signaling. Now, the authors also used PV-Cre mice to ablate a floxed allele of Shk2 specifically in parvalbumin neurons. These mice are also reported to display ASD-like social deficits. The authors should repeat the mGluR5-PAM strategy used in their prior paper to test whether this also works in the PV-Cre/Shank2flox mice. Barnes et al. (Mol Psychiatry 20, 1161-1172, 2015) showed mGluR5 receptors are present on PV neurons and functionally important. Since there are NMDAR on PV neurons, their PAM strategy should also work in the PV-Cre/Shank2flox mice.

→ We appreciate this suggestion but, due to the large amount of additional experiments

that we had to perform during the last 4 months or so, we could not perform the suggested interesting experiment. This was also kindly agreed by the handling editor. Reviewer’s understanding would be much appreciated.

10) In their Introduction, page 4 lines 82-84, the authors state: “Whether Pv neurons display burst firing and how Pv neuronal bursts are generated and regulated are not known”. However, this is technically incorrect. A prior study by Natlow et al. (Neuron, Vol. 38, 805–817, June 5, 2003) that these authors should cite found that Pv neurons in layer 2/3 of the frontal and somatosensory cortex are multipolar bursting cells. Importantly, this paper showed that gap junction coupling between PV neurons could be increased by carbachol (a muscarinic agonist). This suggests that the NMDAR-dependent uncoupling of PV neurons is not developmentally irreversible but can be modulated in adult brain by muscarinic receptor signaling. Since carbachol increases gap junction coupling, would treating WT mice with carbachol produce similar social interaction deficits in WT mice as in Shank2 KO mice?

→ We appreciate the important information. In response, we measured social interaction in WT mice directly infused with carbachol in the mPFC and found that there is no difference in three-chamber social interaction between carbachol-untreated and -treated mice (see below).

Although the effect of carbochol treatment on social interaction is a very interesting topic and surely need further investigations, we think that the abovementioned results are rather preliminary to show in the revised manuscript and make safe conclusions. We thus decided not to show these data in the revised manuscript; reviewer’s understanding would be appreciated. However, we did cite and comment on this important reference (carbachol facilitating gap junctional coupling) in the Introduction.

11) If increased gap junction coupling between PV neurons in PV-Shank2 KO mice contributes to impaired social interaction, how can optogenetic activation of PV neurons rescue social interaction deficits in Figure 7? Won’t Optogenetic activation of PV neurons in mutant mice further amplified the burst firing of PV neurons and further

impair social interaction?

→ To address this comment, we first refined our hypothesis as follows. It is possible that the increased gap junctional activity in *Shank2*^{-/-} Pv neurons might represent a compensatory effort to produce burst firing more strongly upon Pv neuronal stimulation during social interaction, which, however, failed to rescue social interaction under physiological conditions but could do so in the presence of optogenetic 10-Hz Pv neuronal stimulation. Alternatively, the increased gap junctional activity might function as a pathological mechanism that are actively engaged in disrupting mPFC neuronal activity and social cognition, resulting in impaired social interaction.

To differentiate between these two possibilities and also by the suggestion of other reviewers, we attempted connexin-36 knockdown restricted to Pv neurons in the mPFC and found that that it 'increased' social interaction in WT mice (**Fig. 8d**), suggesting that gap junctional activity by itself negatively regulates social interaction. In addition, we attempted a new analysis for the spike-wave synchrony (spike timing in the waves of local field potentials) known to regulate cognitive brain functions. We found that the spike-wave synchrony is strongly reduced by 10-Hz Pv neuronal stimulation (**Fig. 8a,b**), suggesting that burst firing may break Pv neuronal synchrony. These results suggest that the increased gap junctional activity may actively suppress social interaction, whereas optogenetically-induced Pv neuronal bursts inhibit excessive synchrony among Pv neurons. These aspects were clarified in Results and Discussion.

Additional points:

1) Page 4, line 93, please insert the number of WT and *Shank2*^{+/-} mice implanted with tetrodes including the coordinate used.

→ Information inserted (5 mice; AP +1.7 mm, ML ± 0.3 mm).

2) Page 4, line 94 or (Fig 1a), what are dissociation intervals by translocation from reset box to exploration apparatus and so on for each social interaction task. There were two repetitive 10-minutes S-O tests in the same paradigm. How are they different? Were data averaged or pooled for these two tests? Was there any adaptive rundown or desensitization for these repeat measurements? Were any other target pairs set for tests, e.g. E-O or S-E? If yes, name them all.

→ There was no time delay between the rest period (5 min) and the empty/target sessions (E-E, first S-O, and second S-O); now clarified in Methods. The only difference between the first and second S-O sessions was the switching of the S and O targets.

Regarding adaptive rundown, social exploration became weak in the second S-O session in WT mice, which is now clarified in **Supplementary Fig. 1c**. However, *Shank2*-KO mice continued to show impaired social preference in both first and second sessions. The decrease in social exploration in WT mice could be attributable to reduced social drive induced by social habituation. However, discriminative neuronal firings for the S-O target pair were greater than those for the E-E pair in both the first and second S-O sessions relative to (**Supplementary Fig. 1d**). In addition, our previous study on WT mPFC neurons reported positive discriminative neuronal activities in both

the first and second S-O sessions (Lee et al., J Neurosci 36:6926, 2016). We thus pooled the recording data from the first and second S-O sessions. We clarified this in the revised main text.

3) page 25, line 584, what does L-R represent? Also please label the pie charts in Fig 1d.

→ L and R represent left and right, respectively; we clarified this in the main text and figure legend. The numbers inside the pie graphs actually indicate the total number of target pairs that a single neuron can discriminate; we clarified it in the figure legend.

4) Fig 3d, are sEPSC frequency and amplitude different in Pv neurons between WT and Shank2 -/- as in Fig3a? It is interesting to know since Shank2 -/- mice are implicated with NMDAR hypofunction in Fig6a and b.

→ We now show the results of sEPSCs in Shank2-KO mice in **Fig. 3d**, which shows that there is no genotype difference.

5) It seems that the vehicle itself simulated burst and tonic firings, while mefloquine (MQ) suppresses them in WT and KO treated mice (Fig 5b) compared to their non-treated mice (Fig 4b). Two-way ANOVA might be the proper analysis to evaluate burst proportion and firing rates for interaction of gene (WT vs KO) and MQ effects. The same goes for Fig 6c to evaluate the interaction between the gene and the D-cycloserine (DCS) effects by two-way ANOVA.

→ Regarding the notion that vehicle treatment stimulates burst and tonic firings seems to have come from the reviewer's observation that the proportions of burst and tonic firings are different between the data shown in **Fig. 4b** and **Fig. 5b**, which is likely due to the difference in the numbers in the two figure panels (n = 47 and 17 for naïve WT and EGFP-expressing WT, respectively). We thus performed additional experiments for **Fig. 5b** and increased the n number from 17 to 29. The results in the current **Fig. 5b** indicate that burst proportions are comparable to those in **Fig. 4b**.

We also performed two-way ANOVA analysis for firing rates to properly determine genotype x treatment (vehicle/mefloquine) interactions and perform multiple comparisons. The results indicate that mefloquine treatment decreases firing rates in both WT and Shank2-KO mice (**Fig. 5b**). Two-way ANOVA could not be performed for burst/tonic proportions because they were the ratio results in which different neurons were assigned to distinct classes of neuronal firing (no-fire, tonic, and burst).

Regarding two-way ANOVA for DCS effects, we similarly revised the figure panels for firing rates but not for burst proportions (**Fig. 6c**).

6) Fig 7h, the conclusion of boosting Pv neuronal burst by 10-Hz light stimulation that improves social representation and behavioral social interaction is relatively weak. The significance for rescue is only shown in 1 of 5 paired bars whereas it was robust for the phenotype in KO (4/5).

→ We agree with the reviewer and made it clear that the rescue effects for cortical neural representation are modest in the Results and figure legend. However, we have to point out that, in the rescue experiment, the baseline difference was also small (2 of 5 paired bars), which is likely attributable to the factors mentioned in our response to your review comment 1 such as the difference in the genetic background and virus-mediated gene delivery.

Reviewer #2 (Remarks to the Author):

The manuscript “Excitatory synapses and gap junctions cooperate to improve Pv neuronal burst firing and cortical social cognition in Shank2-mutant mice”, by Lee et al, proposes an interesting link between firing properties of networked Pv-positive interneurons, which being modulated by NMDA excitatory drive and gap junction connectivity, drive social behavioral dysfunction in a popular model of autism spectrum disorders. The results are novel and reinforce recent evidence on the literature on the effect of interneuron modulation of social behavior. The authors use advanced methodological tools, including genetics manipulation, optogenetics, in vivo neuronal activity recording and whole-cell patch clamping.

The manuscript will be of interest to the broad field of neuroscience as it proposes links on critical aspects of circuit properties to behavioral modulation.

However, there are several issues with data organization, presentation, clarity, as well as a few logical leaps that prevent a strong endorsement of the manuscript in its present form.

→ We sincerely appreciate the thoughtful comments of the reviewer.

Major issues:

A) L103: The authors propose that neurons may show discriminative responses to S, O and E ”(Fig. 1b [right]), while others showed no target-specific firing”. However, the indicated graph shows a clear target-specific firing towards the Object (O), with no significant response to E or S. Is this just a poorly chosen example? Also, since both target-specific and broadly-tuned neurons may be found in both genotypes, please show examples of both classes for each genotype to avoid confusion.

→ We now show both target-specific and broadly-tuned neurons for both WT and Shank2-KO mice (**Fig. 1b**).

B) L104-109 and Figure 1c-d:

I) The authors propose that KO “neurons responded more strongly to the non-social targets”. The way the data is presented is hard to interpret since the text, figure and figure legend are inconsistent in their labelling and what is being represented is not clear. For example, Fig 1c mentions, Target, Side, S-spc, O-spc, E-spc... These abbreviations are not mentioned in the text or figure legend.

→ Our apologies for the inconvenience. We now provide detailed and consistent descriptions and labelings in the main text, figure legends, and figure panels. In addition, to help readers better understand the data, we moved the Venn Diagram in the original **Supplementary Fig. 3** to current **Fig. 1c**, and show shaded miniature Venn diagrams in **Fig. 1d** to make it clear how the neuronal proportions were derived.

II) Additionally, the figure legend and text, indicate that the total of target-specific neurons is greater in the KO animals. However, eyeballing the sum of “Side + S-spc + O-spc + E-spc” shows a different number than the percentage indicated under “Target”, which is presumably the “total (all combined), targets” as indicated in the Figure legend (L581).

→ Again, we now show the Venn diagram plot in **Fig. 1c** and show miniature Venn diagrams with regions of interest indicated by shades in **Fig. 1d**.

III) Finally, if the authors can clarify the issues above, still, the assertion in L109 that “non-social targets are more strongly represented”, should be perhaps interpreted as non-social discriminative neurons being more prevalent in *Shank2*^{-/-} mPFC. However, a major caveat is if *Shank2* KO mice engage in fewer bouts of social interaction, could it be that there is an undersampling regarding the instances of social nose-pokes in the KO, mice? Can the authors assuage this concern?

→ We thank for the suggestion and agree with the use of “non-social discriminative neurons” rather than “non-social-target neurons” and corrected the text.

Regarding the possibility of undersampling, the numbers of trials for E/S/O targets are comparable in WT and *Shank2*-KO mice, which is now clearly described in the main text as follows and also, in details, in Methods: “These differences are less likely to be attributable to undersampling in *Shank2*^{-/-} mice because the total numbers of nose-poke trials for E-E, S-O, and O-S sessions were comparable in WT and *Shank2*^{-/-} mice (genotype $p = 0.248$; two-way ANOVA; see Methods for further details).”.

IV) It would be useful to indicate the total number of neurons recorded and the number of total trials performed.

→ We now show these numbers (neuron and trial numbers) in **Methods** in details.

V) Figure 1d is also a problem. The point the authors are making is that there is an increase in the total number of targets a single KO neuron responds to. However, the data is very hard to interpret.

a. The figure legend mentions 0-4 pairs (i.e., 5 conditions) but then mentions only 4 pairs (S-O; S-E; E-O and L-R).

→ We made this clear in the main text and figure legend. Briefly, a single neuron gets a zero point when it is not involved in discriminating any of the four target pairs (S-O; S-E; E-O and L-R/left-right), and it gets a full score of four when it discriminates all four target

pairs.

b. More problematic is also the reduction of complexity to dyadic pairs. Presumably, since one neuron is recorded from multiple sessions of different S, E, O combinations, a more refined analysis is necessary to extract meaningful information, e.g. network, graph analysis, or correlation matrixes. Some of this analysis is tentatively exposed in Supplementary Figure 2-3, but the cursory description of these data is problematic.

→ We appreciate this suggestion and have attempted some other ways (i.e. correlation matrix analysis) to analyze the neuronal firings. However, we found that it was not as informative as those shown in **Fig. 1d,e**, **Supplementary Fig. 2**, and **Supplementary Fig. 3** (now in **Fig. 1c**). We have to point out that this analysis can efficiently extract out 'target-discriminative' as well as 'target-selective' neurons using two-way ANOVA analysis of neuronal firings and Venn diagram analyses, compared with the analysis of relatively simple 'target-responding' neurons. This method has also been shown to be useful in analyzing cortical neural responses in WT mice engaged in social interaction (Lee et al., J Neurosci 36: 6926, 2016). We thus described the analysis more carefully in the main text and figure legends and elaborated the figure panels in the revised manuscript. Reviewer's understanding would be much appreciated.

C) Pv vs SOM

a. A major issue in the author's narrative arrives when confronting Figure 1e-h, Figure 2 and Supplementary Figure 6. Specifically, the authors find differences in burst firing on putative inhibitory neurons (Fig 1), go on to observe abnormal oscillations in Shank2 KO mice mPFC and finally segue into Pv neurons as likely culprits. However, since Pv neurons are only one of the populations of inhibitory neurons in the mPFC this raises significant concerns. Particularly because the authors' data show a very small number of PV neurons <5 cells/mm² (Sup. Figure 6), when compared to SOM neurons (10x more prevalent?!) (see <https://doi.org/10.1002/dneu.20853>). If Sup. Fig 6 is correct, what is the likelihood that the phenotype described in Fig 1g-h is derived from Pv-neuron population?

→ We appreciate the careful comment and the useful reference indicating that ~40% and ~30% of cortical interneurons types are Pv- and Somatostatin -positive (Som) neurons, respectively (Rudy et al. 71:45, 2011). The number of interneuron subtypes determined by immunohistochemical antibody staining shown in **Supplementary Fig. 6** (now **Supplementary Fig. 5**) could be misleading because each antibody could have different efficiency of revealing target neurons. We clarified this possibility in the figure legend.

Regarding the possibility of other interneuron types such as Som neurons play certain roles in the results shown in **Fig. 1f-i**, we think the spike shapes shown in **Fig. 1f** is important, where we defined interneurons as those displaying peak-to-valley ratio lower than 1.5 and half-valley width smaller than 250 μ sec, which could contain Som neurons in addition to Pv neurons. An efficient way of differentiating Pv and Som neurons is to try optotagging combined with in vivo recording, and a previous report (Kim et al. Neuron 92:902, 2016) has shown that ~70% and ~30% of the putative

interneurons are Pv and Som neurons, respectively. We thus cannot exclude the possibility that Som neurons, in addition to Pv neurons, contribute to the results observed in **Fig. 1f-i**. We clarified this in Results. However, we have to point out that all the experiments following this result was performed using labeled Pv neurons.

b. Can the authors differentiate the inhibitory neurons recorded in vivo segregated as Pv-like or SOM-like neurons?

(see <https://doi.org/10.1126/sciadv.aay4073> and <https://doi.org/10.1016/j.cell.2015.11.038>)

→ Unfortunately, our experimental design was to use naïve and Shank2-KO mice in which mPFC Pv/Som neurons were not differentially tagged, which is certainly an interesting direction for future studies. Again, we toned down our claims and clearly stated in Results that the changes observed in the putative interneurons could involve both Pv and Som neuronal dysfunctions, with the citation of the references on the contribution of multiple interneuronal cell types to social/cognitive functions.

c. Note that the reference indicated in that section (Lim et al. 2018), presumably DOI:10.1016/j.neuron.2018.10.009, is not indicated in the reference list.

→ We appreciate this comment. It is now included.

D) Regarding the optogenetic stimulation of Pv neurons (Fig 4) the authors find an increase in burst firing when neighboring Pv-Pv linked-neurons are recruited by ChR2 activation. However, a competing hypothesis, is that the rheobase, AP threshold or resting membrane potential of the target cells is already somewhat perturbed due to loss of Shank3. Can the authors provide a table with the membrane properties of the recorded neurons to assuage these concerns?

→ In response, we now provide a table showing that there is no genotype difference in parameters of intrinsic neuronal excitability, including rheobase, AP threshold, and AHP, in both pyramidal and Pv neurons (**Supplementary Table 2**). One small exception was an increase in the input resistance in Shank2-KO Pv neurons, although the current-firing curve, a parameter more important for neuronal excitability during activity situations, is not altered in both pyramidal and Pv neurons (**Fig. 3c,f**).

E) Gap junction and electrical coupling:

a. L228: The authors indicate that gap junctions may function has a low-pass filters to explain the difference between 10 and 40Hz stimulation... However, a common understanding is that this filtering capability is significantly noticeable only when comparing between very high-frequency events (spikes) but perhaps not has much within lower frequency (10-100Hz). Additional experiments should be formed to assess the attenuation in the neighboring neurons using a wider range of frequencies and ChETA or direct current injections.

→ In response, we repeated the experiments at three different stimulation frequencies

(10, 40, and 80 Hz), which was possible with ChR2 stimulation in our experimental design of giving brief stimulations. The results indicated that burst firing is efficiently elicited by 10-Hz but not 40- or 80-Hz stimulation in neighboring Pv neurons (**Supplementary Fig. 8**).

b. Another factor pertaining to gap junctions will be a charge diffusion that is more greatly dispersed in more connected cells (in this case the Wt), when compared to less connected cells (Shank2 KO), by virtue of the reduced number of Pv partners in the network of the KOs (Supplementary Figure 6). Can the authors expand their modeling work to estimate the impact of a reduced number of electrically coupled partners or discount the consequences of having a system of similarly expressed gap-junctions (no upregulation) but where the lower partner connectivity putatively leads to greater charger transfer per gap junction.

→ In response and given the result that Pv neuronal density ‘tends’ to be decreased in Shank2-KO mice (not actually reduced), we performed revised model simulations in which the total number of Pv neurons in the network was decreased from $n = 100$ to 50 while the total number of gap junctions (GJs) and the strength of individual GJs were unaltered. We found that burst firing (as shown by burst ratio and burst-firing spike number) in the network is increased when Pv neuronal number in the network is reduced to 50, as compared with the original number of 100, while burst firing showed the same tendency of increase when GJ number and strength were increased (**Supplementary Fig. 9c,d**). This result suggests that reduced Pv neuronal density with unaltered total number of GJs and strength of individual GJs can increase burst firing in the network. Therefore, it is possible that the decreasing tendency of cortical Pv neuronal density in Shank2-KO mice (as shown by the immunohistochemistry results) may promote burst firing in the mutant PV neuronal network. We commented on these aspects in Results.

F) There is a large discrepancy in the firing properties of Wt neurons recorded in Fig 4b (control) and 5d (Vehicle condition) or Fig 6c. The same between Fig 4a and Fig7d (here the KO stimulated has a very different response even in the trace example!). Should these be equivalent preparations in vehicle and control? If yes, what can explain such a significant difference in firing properties (tonic, burst etc), the average firing rate of the neurons, or traces?

→ Regarding the difference between the results in **Fig. 4b** and **Fig. 5b/Fig. 6c** (the proportions of burst and tonic firings), it is likely due to the difference in the n numbers in the two figure panels ($n = 47$ and $17/20$, respectively). We thus performed additional experiments for **Fig. 5b/Fig. 6c** and increased the n numbers from $12/20$ to $29/43$. The results now indicate that the proportions of the burst and tonic firings are comparable in the two figures (**Fig. 4b** and **Fig. 5b/Fig. 6c**).

Regarding the difference between **Fig. 4a** and **Fig. 7d** (actually **Fig. 6d**; now in **Supplementary Fig. 11**), the n numbers of neurons are 42 and 20, respectively. However, we could not perform additional experiments for **Supplementary Fig. 11** for the large volume of electrophysiological experiments that we had to perform during the

revision (i.e., shown in **Fig. 3d**, **Fig. 5b**, **Fig. 6c**, **Fig. 6d**, and **Supplementary Fig. 6a,b**); reviewer's understanding would be much appreciated. We moved the results in old **Fig. 6d**, showing the lack of the effects of DCS in directly stimulated WT and KO neurons, to **Supplementary Fig. 11**.

G) Finally, a major weakness of the work is that the very interesting assertion that gap junctions are playing a causative role in the bursting activity deficits in Shank2 KO mice, which in turn would lead to social behavioral deficits; falls short in a direct mechanistic test. One of two experiments would be critical to perform:

- a. Further enhance the upregulation of gap junctions (connexin 36?) in a Pv-specific manner to support the authors' theory of a compensatory role of this mechanism, thereby expecting a rescue of the phenotype.
- b. Or, selectively perturb Gap junctions in Pv neurons in wildtype mice to phenocopy social behavioral dysfunction.

→ To address this comment, we first refined our hypothesis as follows. It is possible that the increased gap junctional activity in *Shank2*^{-/-} Pv neurons might represent a compensatory effort to produce burst firing more strongly upon Pv neuronal stimulation during social interaction, which, however, failed to rescue social interaction under physiological conditions but could do so in the presence of optogenetic 10-Hz Pv neuronal stimulation. Alternatively, the increased gap junctional activity might function as a pathological mechanism that are actively engaged in disrupting mPFC neuronal activity and social cognition, resulting in impaired social interaction.

Regarding specific revision experiments, we thought about the option 'a' and but discouraged to perform the suggested experiment because connexin 36 overexpression surpassing the endogenous levels might induce some dominant negative effects. We thus attempted the second suggested experiment. Specifically, we tried to perturb gap junctional activity by 1) direct mefloquine infusion into the mPFC and 2) AAV-mediated knockdown of connexin-36 restricted to Pv neurons in the mPFC.

In the first experiment (direct mefloquine infusion), we found that this treatment does not affect social interaction in WT or Shank2-KO mice (**Fig. 8c**). However, this could be attributable to that gap junctions are detected in various GABAergic cell types, including fast-spiking Pv neurons, Som neurons, and polar burst cells (a subtype of Pv neurons distinct from fast-spiking Pv neurons); clarified in the main text.

In the second experiment (connexin-36 knockdown), we attempted Pv neuron-specific knockdown of connexin-36 using two independent AAV knockdown constructs. Pv neuron-specific connexin-36 knockdown by AAV infection in the mPFC in WT mice induced 'increased' social interaction in the three-chamber test (**Fig. 8d**). This suggests that gap junctional activity by itself has negative influences on social interaction, and that the increase gap junctional activity in Shank2-KO mice that failed to rescue social interaction in the absence of Pv neuronal stimulation may also actively inhibit normal social interaction.

In addition, we attempted a new analysis of the spike-wave synchrony, known to regulate various cognitive functions including attention, cognitive flexibility, and social cognition. Here, we found that the 10-Hz Pv neuronal stimulation suppresses spike-

wave synchrony in the Shank2-KO mPFC in many frequency ranges, including theta and gamma, during social interaction (**Fig. 8a,b**). These results suggest that increased gap junctional activity in Shank2-KO mice abnormally increases Pv neuronal synchrony and thus suppress social interaction, and that 10-Hz stimulation-induced Pv neuronal bursts may suppress the excessive spike-wave synchrony. These results were added to Results and commented in Discussion.

Minor points:

Figure 1a –change 2nd S-O to O-S to avoid confusion.

→ We changed the label in the figure.

Confirm the statistical analysis is Figure 2c, since a “eyeball analysis” (i.e. close means with SEM overlap) raises some concerns on the comparison between Pre and Post Gamma PSD in the Wt animals showing $p < 0.05$.

→ We confirmed that the results in Fig. 2c are statistically significant likely because of the high n numbers (n = 52 for WT).

Why were only female mice used in the patch experiments? Are the authors confident there are no differences between genders? Can they provide data to this point? This is particularly important in a study that goes from cellular, circuit and behavioral analysis and where a sense of continuum is given.

→ We actually used female mice only for the results shown in **Fig. 3f-j** (now **Fig. 3h-k**), and all other patch experiments were performed using male mice, meaning that only a small fraction of the electrophysiology experiments were performed in females. We could not try an additional experiment using males to replace the data in **Fig. 3h-k**, for other experiments and the lack of sex difference in the behaviors of Shank2-KO mice (Won et al., Nature 486:261-265, 2012). We clarified this by changing the text in Methods as follows: “Heterozygous Pv-Cre male 30 mice over the age of 12 weeks were used for all the experiments except for the patch experiment with optogenetic stimulation, where we used female mice.” was changed to “Heterozygous Pv-Cre male 30 mice over the age of 12 weeks were used for all the experiments except for the patch experiment shown in Fig. 3h-k, where we used female mice, partly based on the previous report that male and female Shank2-KO mice show comparable levels of behavioral abnormalities¹.”.

L576 Figure 1 – Symbol is missing.

→ We appreciate the careful reading; the missing symbol (σ) was added.

Figure 1b Wt graph is repeated in Supplementary figure 1b WT.

→ We appreciate the careful reading. We now show a different example in **Fig. 1b**.

Reviewer #3 (Remarks to the Author):

Review of Lee et al Nat Comms 2021:

This manuscript by Lee et al. uses in-vitro and in-vivo electrophysiology together with behavioral testing to investigate the cellular and circuit mechanism of social behavior dysfunction in the Shank2 mouse model of autism. The authors provide compelling evidence for the involvement of both gap junction hyperfunction and NMDA receptor hypofunction in the behavioral phenotypes displayed by Shank2^{-/-} mice. The electrophysiological experiments are well-designed and carefully executed, using a set of paradigms previously developed and validated by the authors. The authors further develop a novel approach for testing the responses of PV neurons to optogenetic stimulation of neighboring PV neurons, showing that differences between the WT and KO mice are prominent in this aspect of cortical physiology. This is an innovative concept which has received little attention in the investigation of the pathophysiological mechanisms of autism. Although the authors do not provide a solid causal link between the gap junction dysfunction and the social behavior phenotypes, this manuscript is exciting for its innovative mechanistic insight, and will be of interest to a broad community of scientists studying the prefrontal cortex, autism spectrum disorders and synaptic physiology.

→ We sincerely appreciate the thoughtful comments of the reviewer.

Major comments:

- Can the differences in burst firing across targets in Fig. 1h (in WT mice) be attributed merely to the differences in firing frequencies between the targets (S vs. R)? Although the average firing rate was not different in the inhibitory neurons (Fig. 1f) it would be good if the authors could show that shuffling the spike times eliminates the burst firing differences.

→ In response, we shuffled the spike times and found that it eliminates the burst firing differences (**Supplementary Fig. 3e**).

- Figures 4, 5 and 6 rely on a novel technique, in which the authors stimulate individual PV neurons while recording from the stimulated cell or a neighboring one. This experiment is very interesting, and it allowed the discovery of a novel gap junction phenotype in the Shank2 KO mice. This is a very clever experimental paradigm. However, some technical information is missing in the description of this experiment. For instance, what was the light power density used for these experiments? What was the illuminated area when using the DMD device? Are the authors confident that they are really stimulating only a single PV neuron? A control experiment for this could be recording in the same configuration with GABA and glutamate synaptic blockers. This

will allow quantification of the residual photocurrent triggered directly vs. the gap junction-mediated current. At minimum, the authors should report the distances between the stimulated and recorded neurons and show that these distances are not systematically different between the two genotypes.

→ We now provide technical details for single Pv neuronal stimulation using Polygon DMD pattern illuminator (Mightex) in Methods, including the light power, the illuminated area, and how we tried to ensure to select and stimulate a well-isolated single Pv neuron.

In addition, we performed an additional experiment, in which light-induced burst firing was measured in the presence of synaptic blockers (NBQX for AMPARs, APV for NMDARs, and GABAzine for GABA receptors). We found that the difference between WT and *Shank2*^{-/-} Pv neurons was moderately affected by the synaptic blockade, with no effect of synaptic blockade in mean-firing rates in both WT and *Shank2*^{-/-} Pv neurons and with a decrease burst firing and an increase in tonic firing only in *Shank2*^{-/-} (not WT) Pv neurons (**Supplementary Fig. 6b**). This moderate (~2-fold) decrease in burst firing by synaptic blockade in *Shank2*^{-/-} Pv neurons contrasts with the a much stronger (~20-fold; from 16% to 0.67%) mefloquine-dependent decrease in burst firing in *Shank2*^{-/-} Pv neurons (**Fig. 5b**), suggesting that the majority of the burst firing in the *Shank2*^{-/-} Pv neurons are mediated by gap junctions rather than synaptic junctions. We clarified this in the Results.

- The authors demonstrate that connexin-36 expression is not different between WT and KO mice across a wide range of ages. The fact that PV neurons are slightly less abundant still cannot account for the substantial increase in gap-junction mediated currents in the KO mice. Another potential explanation would be that the arborization of PV neurons in the KO mice is more extensive and this allows for more extensive coupling between cells. Anatomical tracing of dye-filled PV neurons would reveal whether structural differences might underlie the differences in electrophysiology identified by the authors.

→ We appreciate this idea. We performed the suggested experiment measuring arborization and found that there is no genotype difference in the dendritic arborization pattern, as determined by Sholl analysis (**Supplementary Fig. 10**).

- Figure 7: the authors state that PV neuron stimulation “restored social interaction in *Shank2*^{-/-} mice”. However the statistical comparison in Fig. 7 is unclear - is there a significant across-group interaction in the 2-way ANOVA? The indication on the figure only implies a within-group repeated measures effect.

→ We now show the two-way ANOVA results in **Supplementary Fig. 12**, which shows that social interaction, but not object interaction, is increased in the KO-ChR2 group but not in the KO-EGFP, KO-ChETA or KO-SSFO groups.

- The authors rightfully claim in their Discussion that their results might indicate an involvement of mPFC gap junction hyperfunction in the behavioral phenotypes observed

in *Shank2*^{-/-} mice. I wonder whether they have tried to directly test this hypothesis by applying mefloquine directly in the mPFC in the KO mice? This might indeed be a direct test of their hypothesis and help consolidate the novel concept introduced in this manuscript, that gap junction hyperfunction is a potential mechanism of autism-related dysfunction and a potential target for future therapies.

→ To address this comment, we first refined our hypothesis as follows. It is possible that the increased gap junctional activity in *Shank2*^{-/-} Pv neurons might represent a compensatory effort to produce burst firing more strongly upon Pv neuronal stimulation during social interaction, which, however, failed to rescue social interaction under physiological conditions but could do so in the presence of optogenetic 10-Hz Pv neuronal stimulation. Alternatively, the increased gap junctional activity might function as a pathological mechanism that are actively engaged in disrupting mPFC neuronal activity and social cognition, resulting in impaired social interaction.

In the revision experiments, we attempted to perturb gap junctional activity in the mPFC by 1) direct mefloquine infusion and 2) knockdown of Cx-36 restricted to Pv neurons in the mPFC. In the first experiment (direct infusion of mefloquine to the mPFC of WT and *Shank2*-KO mice), we found that this treatment does not affect social preference in WT or *Shank2*-KO mice (**Fig. 8c**). However, this could be attributable to that gap junctions are detected in various GABAergic cell types, including fast-spiking Pv neurons, somatostatin-positive neurons, and polar burst cells (a subtype of Pv neurons distinct from fast-spiking Pv neurons); clarified in the main text.

In the second experiment (connexin-36 knockdown experiments), we attempted Pv neuron-specific knockdown of connexin-36 using two independent AAV knockdown constructs. WT mice with Pv neuron-specific connexin-36 knockdown by AAV infection in the mPFC showed ‘increased’ social interaction in the three-chamber test (**Fig. 8d**). This suggests that gap junctional activity by itself has negative influences on social interaction, and that the increased gap junctional activity in *Shank2*-KO mice that failed to rescue social interaction in the absence of Pv neuronal stimulation may also actively inhibit normal social interaction.

In addition, we attempted a new analysis of the spike-wave synchrony, known to regulate various cognitive functions including attention, cognitive flexibility, and social cognition. Here, we found that the 10-Hz Pv neuronal stimulation suppresses spike-wave synchrony in the *Shank2*-KO mPFC in many frequency ranges, including theta and gamma, during social interaction (**Fig. 8a,b**). These results suggest that the increased gap junctional activity in *Shank2*-KO mice promotes Pv neuronal synchrony and thus suppress social interaction, and that the Pv neuronal burst induced by 10-Hz stimulation may suppress the spike-wave synchrony in *Shank2*-KO mice to promote social cognition and interaction. These aspects were clarified in Results and Discussion.

Minor comments:

- Line 85 - “cognitions” should be singular.

→ We corrected it.

- Lines 88-91 - sentence with multiple clauses - very difficult to read. Please simplify and avoid multiple clauses throughout the manuscript.

→ We tried to minimize clauses throughout the manuscript.

- 595 firing-firing rate?

→ We corrected it.

- Supplementary Fig. 4: instead of “criteria”, the term should perhaps be “threshold”.

→ We corrected it.

- Line 149 - it seems that theta power did decrease in the mPFC upon nose poke.

→ We appreciate, and it is now described.

- General comment: the authors use the term “nose poke” to describe interaction with the target. This term is traditionally used in operant conditioning to describe a specific action made by the animal. The “nose poke” in that context has a very defined motor pattern and is not similar to the sniffing that mice perform when investigating a conspecific or an object. Perhaps a better term would be “sniffing onset”?

→ We appreciate the suggestion. In response, we changed “nose poke” to “sniffing (nose poke)” to maximize the benefit of both terms. However, “sniffing onset” is slightly different from “nose poke”, which is also used in the manuscript to indicate the whole series of actions of social exploration/interaction, including sniffing/nose-poke onset and maintenance; we tried to properly differentiate ‘sniffing onset’ and ‘sniffing (nose-poke)’ throughout the manuscript.

- Line 167: “PV neuronal density tended to decrease strongly in the Shank2^{-/-} mPFC, unlike that of somatostatin-positive neurons” - the word “strongly” seems inappropriate given that none of the comparison reach significance, while somatostatin interneurons are actually significantly reduced in the Cg. This claim should somehow be toned down.

→ We agree and toned down our description by removing the “strongly”.

- Figure 4: Do PV neurons in Shank2 KO mice have the same ChR2 photocurrent amplitude as WT PV neurons when stimulated directly? The observed changes might be a simple result of altered translation and membrane trafficking in the KO mice.

→ We could observe comparable amounts of photocurrents in WT and Pv neurons in both AP-generating or AP-non-generating conditions (**Supplementary Fig. 6a**). We clarified this in the revised Methods.

- Supplementary information- patch clamp methods: line 121: please check the

temperatures stated.

→ We appreciate and made the change; “3–32 °C” was changed to “30–32 °C”.

Reviewers' Comments:

Reviewer #1:

Remarks to the Author:

The authors have satisfactorily answered most of this reviewer's concerns. Many of our concerns centered around the Shank2flox/flox mouse, but since the authors used the Shank2 null mouse in a PV-Cre mouse to label/express viral vectors in PV neurons, experiments using the Shank2 flox mouse were clearly never carried out. We apologize for the error. The authors did undertake 2 important additional experiments that we suggested: one that has partially clarified the interpretation of their results that they include, and the other that muddies things a little (that they don't include).

We understand that carrying out the experiments with the mGluR5 PAM might not be feasible given time limitations. Nonetheless, the authors should address their own prior published result in their Discussion. Also, the authors should address possible limitations of their study (particularly in light of the carbachol experiment).

Reviewer #2:

Remarks to the Author:

The authors have reasonably addressed this reviewer's concerns, improved clarity, added a substantial amount of data that elevates the quality of the work and provides further mechanistic insight.

Therefore I endorse the publication of the article in its present form.

Reviewer #3:

Remarks to the Author:

The authors have revised the manuscript according to my previous comments, and have done a very good job in addressing the points raised. There is clearly more to be done on this topic, but the study in its present form is suitable for publication in Nature Communications. I have no further comments and wish to congratulate the authors on a very nice study.

Point-by-point response to reviewers' comments

REVIEWER COMMENTS

Reviewer #1 (Remarks to the Author):

The authors have satisfactorily answered most of this reviewer's concerns. Many of our concerns centered around the Shank2flox/flox mouse, but since the authors used the Shank2 null mouse in a PV-Cre mouse to label/express viral vectors in PV neurons, experiments using the Shank2 flox mouse were clearly never carried out. We apologize for the error. The authors did undertake 2 important additional experiments that we suggested: one that has partially clarified the interpretation of their results that they include, and the other that muddies things a little (that they don't include).

We understand that carrying out the experiments with the mGluR5 PAM might not be feasible given time limitations. Nonetheless, the authors should address their own prior published result in their Discussion. Also, the authors should address possible limitations of their study (particularly in light of the carbachol experiment).

→ In response, we generated a new paragraph in Discussion commenting on the potential interplays between NMDARs, gap junction, and known Pv neuronal modulators such as mGluR5 and muscarinic receptors as follows:

"These potential interplays between NMDARs and gap junctions in Pv neurons could be further associated with other modulators in Pv neurons, including mGluR5 and muscarinic receptors. Our previous study on *Shank2*^{-/-} mice has shown that activation of metabotropic glutamate receptor 5 by a positive allosteric modulator (CDPPB) rescues NMDAR function and social deficits¹. In addition, Pv neuronal mGluR5 is critical for Pv neuronal development, network activity, and behaviors², suggesting the possibility that mGluR5 positive allosteric modulators may improve Pv neuronal function and social interaction in *Shank2*^{-/-} mice. In addition, given that muscarinic receptor activation can increase gap junctional coupling in adult Pv-positive multipolar bursting cells in frontal and somatosensory cortices (layer 2/3)³, muscarinic receptor agonists such as carbachol may modulate Pv neuronal burst firing and social interaction in *Shank2*^{-/-} mice."

Reviewer #2 (Remarks to the Author):

The authors have reasonably addressed this reviewers concerns, improved clarity, added a substantial amount of data that elevates the quality of the work and provides further mechanistic insight.

Therefore I endorse the publication of the article in its present form.

→ We appreciate the final comment of the reviewer.

Reviewer #3 (Remarks to the Author):

The authors have revised the manuscript according to my previous comments, and have done a very good job in addressing the points raised. There is clearly more to be done on this topic, but the study in its present form is suitable for publication in Nature Communications. I have no further comments and wish to congratulate the authors on a very nice study.

→ We appreciate the final comment of the reviewer.

References

1. Won, H., *et al.* Autistic-like social behaviour in Shank2-mutant mice improved by restoring NMDA receptor function. *Nature* **486**, 261-265 (2012).
2. Barnes, S.A., *et al.* Disruption of mGluR5 in parvalbumin-positive interneurons induces core features of neurodevelopmental disorders. *Mol Psychiatry* **20**, 1161-1172 (2015).
3. Blatow, M., *et al.* A novel network of multipolar bursting interneurons generates theta frequency oscillations in neocortex. *Neuron* **38**, 805-817 (2003).